# Sulfur starvation-induced autophagy in *Saccharomyces cerevisiae* involves SAM-dependent signaling and transcription activator Met4

Magali Prigent[1,2], Hélène Jean-Jacques[1], Delphine Naquin [1], Stéphane Chédin[1], Marie-Hélène Cuif[1,2], Renaud Legouis [1,2] & Laurent Kuras [1] ✉

Autophagy is a key lysosomal degradative mechanism allowing a prosurvival response to stresses, especially nutrient starvation. Here we investigate the mechanism of autophagy induction in response to sulfur starvation in *Saccharomyces cerevisiae*. We found that sulfur deprivation leads to rapid and widespread transcriptional induction of autophagy-related (*ATG*) genes in ways not seen under nitrogen starvation. This distinctive response depends mainly on the transcription activator of sulfur metabolism Met4. Consistently, Met4 is essential for autophagy under sulfur starvation. Depletion of either cysteine, methionine or SAM induces autophagy flux. However, only SAM depletion can trigger strong transcriptional induction of *ATG* genes and a fully functional autophagic response. Furthermore, combined inactivation of Met4 and Atg1 causes a dramatic decrease in cell survival under sulfur starvation, highlighting the interplay between sulfur metabolism and autophagy to maintain cell viability. Thus, we describe a pathway of sulfur starvation-induced autophagy depending on Met4 and involving SAM as signaling sulfur metabolite.

Sulfur is a chemical element critical to life present in a number of indispensable organic compounds, including the two amino acids methionine and cysteine, the ubiquitous methyl-group donor S-adenosylmethionine (SAM) and the antioxidant tripeptide glutathione (GSH, γ-glutamyl-cysteinyl-glycine). In mammals, methionine is an essential amino acid serving as a precursor for the biosynthesis of SAM, cysteine and GSH[1]. The clinical features associated with known rare inherited disorders in sulfur amino acid metabolism underline the importance of the sulfur compounds in human health. For instance, homocystinuria, usually caused by mutations in the biosynthetic pathway leading from methionine to cysteine, manifests itself in the form of symptoms affecting multiple organs/systems, including the eyes, the skeleton, the vascular system, and the central nervous system[2]. Disruption of sulfur amino acid metabolism could cause pathophysiological states through epigenetic dysregulation[3]. In particular, SAM depletion in mouse liver and human colon cancer cells coincides with global changes in histone H3 lysine 4 trimethylation and affects expression of genes involved in specific cellular processes[4,5]. Disruption of sulfur amino acid metabolism can also be beneficial. Methionine restriction obtained through dietary limitation or metabolic mutations extends lifespan in different species including yeast[6–8], worms[9,10], and mammals[11,12]. Although the exact underlying mechanisms are still unknown, several possibilities, including modification of gene expression and induction of antioxidant defense have been proposed[13].

[1]Université Paris-Saclay, CEA, CNRS, Institute for Integrative Biology of the Cell (I2BC), Gif-sur-Yvette, France. [2]INSERM U1280, 91198 Gif-sur-Yvette, France. ✉e-mail: laurent.kuras@i2bc.paris-saclay.fr

Sulfur amino acid metabolism is conserved between yeast and mammals, except that yeast can assimilate inorganic sulfur through the sulfate assimilation pathway[14,15]. Expression of *S. cerevisiae* sulfur amino acid biosynthesis genes (referred to as *MET* genes) are controlled by complex regulatory mechanisms that involve several transcriptional factors, including the transcriptional activator Met4[16–18]. *MET* genes and a number of sulfur compound-specific transporter genes are rapidly and coordinately activated in response to cellular shortage of sulfur amino acids[18].

Autophagy is a key nutrient starvation-protective mechanism, in which cytoplasmic components are recycled through degradation in the lysosome (the vacuole in yeast) to maintain pools of basic biosynthetic precursors[19,20]. Several types of autophagy exist, differing in their mode of substrate delivery to the lysosome/vacuole and their substrate selectivity. Macroautophagy (hereafter referred to as autophagy) involves the sequestering of cytoplasmic material into an isolation membrane, the phagophore, which after expansion and closure generates a double-membrane vesicle called the autophagosome. Fusion of the autophagosome with the lysosome/vacuole leads to degradation of the sequestered material. The process generally ends with the release of breakdown products into the cytosol[21]. Autophagy is also involved in the removal of harmful cytoplasmic constituents (including proteotoxic aggregates, damaged organelles, and pathogens) as well as in various developmental processes; consequently autophagy dysregulation is associated with a number of pathologies such as cancer, neurodegeneration and microbial infection[22].

Autophagy-related (Atg) proteins were first identified in *S. cerevisiae*[23–25]. More than forty Atg proteins have been described, almost all functionally conserved among eukaryotes[26,27]. Autophagosome biogenesis is a multi-step process that necessitates cargo recognition and coordinated transfer of lipids from various reservoirs. The core machinery for autophagosome biogenesis include six main functional entities: the Atg1 (ULK1/2 in mammals) protein kinase complex, Atg9-containing lipid vesicles, the class III phosphatidylinositol 3-kinase (PI3K) complex, and the Atg12 and Atg8 (LC3/GABARAP in mammals) conjugation systems[28–30].

Autophagy regulation takes place at transcriptional and posttranscriptional levels and involves various mechanisms, including epigenetic changes, transcriptional repression and activation, and diverse types of protein modification (phosphorylation ubiquitylation…)[31–33]. In *S. cerevisiae*, phosphorylation of Atg1 complex by target of rapamycin complex (TORC) 1 was shown to impede autophagy initiation in nutrient-rich conditions, so that TORC1 inactivation is viewed as a key trigger promoting autophagy upon starvation[34,35]. Transcription of some *ATG* genes is also upregulated under certain starvation conditions, such as nitrogen starvation[31,32]. However, the significance of the transcriptional regulation of autophagy has not been much explored and key questions remain unanswered, including: What is the exact raison d'être of this particular level of regulation in the response to starvation? How and to what extent is the autophagic process affected by transcriptional changes?

In this report, we focus on the transcriptional regulation of autophagy in response to sulfur starvation. Our results establish Met4 as a main regulator of autophagy ensuring coordination between the metabolic and autophagic responses to sulfur starvation, with SAM serving as key signaling metabolite.

## Results

### Sulfur starvation leads to transcriptional induction of *ATG* genes

Ohsumi and colleagues reported in the early 1990s that sulfur-starved yeast cells accumulate autophagic bodies in the vacuole[36]; however, sulfur starvation-induced autophagy in yeast has not been further studied since then and the underlying mechanisms remain unknown. To monitor the autophagic flux under sulfur starvation, we utilized the GFP-Atg8 processing assay[37], which quantifies the levels of free GFP

moiety resulting from the cleavage of GFP-Atg8 delivered to the vacuole. Starvation was performed as follows: cells were grown to exponential phase in synthetic sulfur-free (SF) medium supplemented with methionine as unique sulfur source, collected by filtration, washed, and then transferred into fresh SF-medium. We observed low GFP-Atg8 expression prior starvation but no free GFP (Fig. 1a). Following sulfur depletion, GFP-Atg8 expression increased significantly and free GFP accumulated gradually (Fig. 1a), demonstrating GFP-Atg8 processing. We also performed live-cell microscopy analysis to follow the cellular localization of GFP fluorescence (Fig. 1b). The results showed an overall increase of cellular fluorescence in the first 2 h after sulfur depletion, accompanied by an increase of the number of GFP foci, and followed by progressive accumulation of GFP into the vacuole. Altogether, these experiments confirmed that sulfur depletion results in autophagy induction.

We then asked whether a defect of autophagy would affect the survival capacity of cells deprived of sulfur. For this, a wild-type (WT) strain and several mutants of core components of the autophagy machinery were submitted to sulfur starvation and the number of cells able to resume growth when placed back on medium containing sulfur was measured over time (Fig. 1c). The number of cells increased slightly during the first day of starvation, presumably because the cells did not stop dividing immediately and contained enough sulfur to complete their cycle. Then this number remained quite stable during the 10-day starvation period in the case of the WT strain, whereas it dropped by more than 10-fold in the case of the autophagy-defective strains. These results demonstrate that autophagy is critical for cell survival under conditions of sulfur starvation.

The increase in GFP-Atg8 expression let us hypothesize a regulation at the transcriptional level. To determine how *ATG* genes respond to sulfur depletion, we performed transcriptional profiling, using RNA-sequencing, before and at several time points after sulfur depletion (Fig. 1d; Supplementary Fig. 1 and Table 5). Out of the 36 *ATG* genes, 16 (44%) had maximal log2-fold change (LFC) > 2 and only one had negative maximal LFC, indicating a quite homogeneous response. For comparison, out of the 6663 ORFs included in our analysis, 11 % had maximal LFC > 2 and 35% had negative maximal LFC while 45% of the 44 genes belonging to the sulfur amino acid biosynthesis pathway had maximal LFC > 2 and 30% had negative maximal LFC (*see* Supplementary Fig. 1a and Table 6). RT-qPCR analysis of selected *ATG* genes gave results in line with the RNA-Seq results (Supplementary Fig. 1c). These analyses demonstrate that the vast majority of *ATG* genes are upregulated following sulfur depletion.

Even though *ATG* genes responded quite homogeneously following sulfur depletion, *ATG41* was induced noticeably more strongly than the others (Fig. 1d). Several reports have presented evidence supporting a role for Atg41 in autophagy, possibly during autophagosome formation, but its molecular function remains unknown[38–40]. To establish further that Atg41 was required for the autophagic response to sulfur depletion, we monitored GFP-Atg8 processing and cellular localization in *atg41Δ* cells. GFP-Atg8 was expressed prior starvation in the *atg41Δ* mutant but expression did not increase following sulfur depletion (Supplementary Fig. 2a). Moreover, only 12% of GFP-Atg8 was processed after 8 hours of starvation in the *atg41Δ* cells (75% in the WT cells), resulting in very low levels of GFP release. Under the microscope, the two strains showed similar proportions of cells with GFP foci; however, the *atg41Δ* mutant did not show accumulation of GFP fluorescence in the vacuole contrary to the WT strain (Supplementary Fig. 2b). We also carried out the Pho8Δ60 assay, which measures the alkaline phosphatase (ALP) activity resulting from Pho8Δ60 delivery to the vacuole through autophagy. In the WT strain, Pho8Δ60 ALP activity was strongly increased in the hours following sulfur depletion (Supplementary Fig. 2c). In contrast, there was almost no increase of activity in the *atg41Δ* strain, similarly to the *atg1Δ* strain. Therefore, *ATG41* inactivation caused an important reduction of the

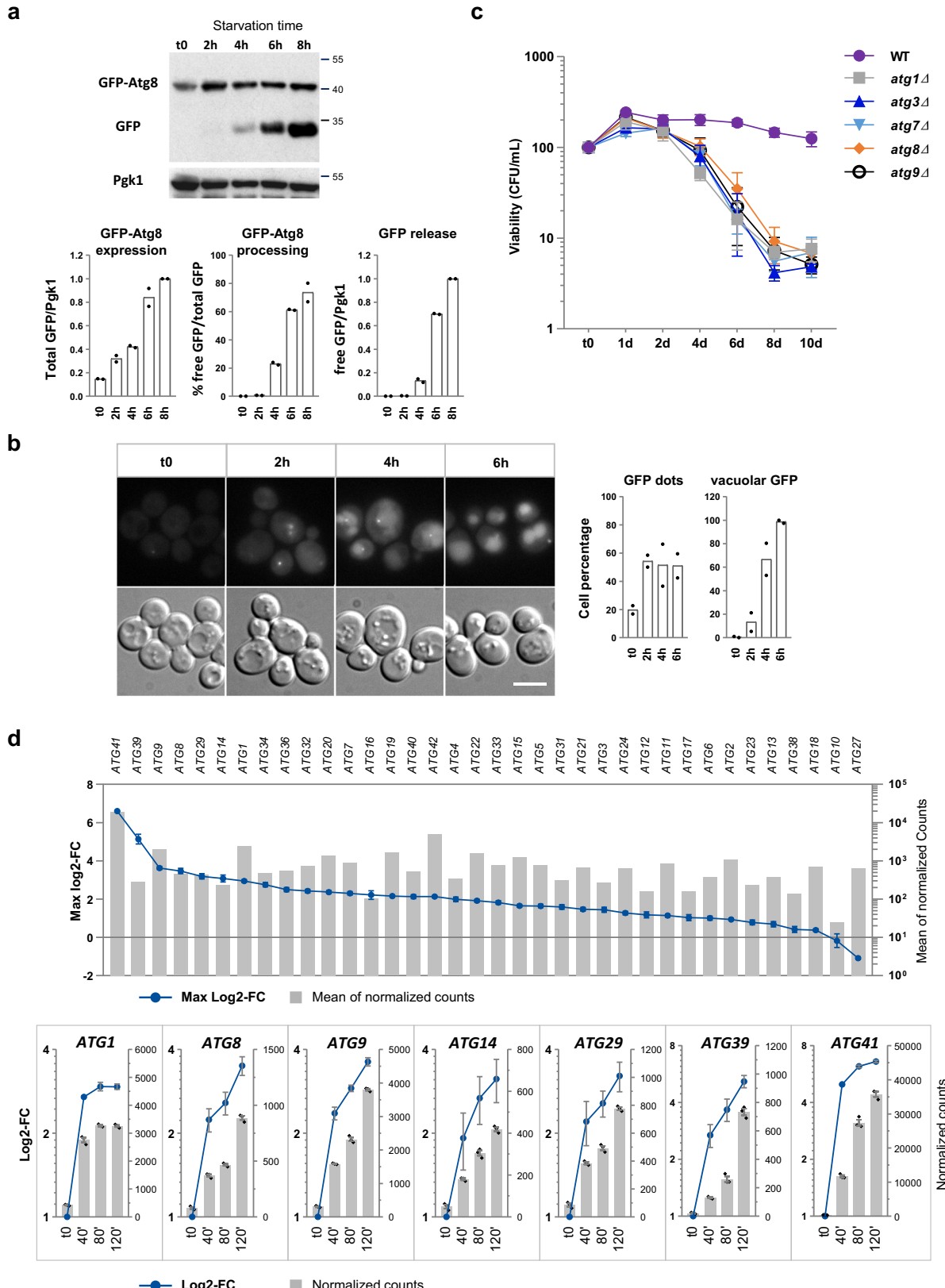

autophagic activity induced by sulfur starvation, thereby demonstrating that Atg41 plays a substantial role in the autophagic response under this starvation condition. The lack of increase of GFP-Atg8 expression in the *atg41Δ* mutant prompted us to carry out transcription analysis (Supplementary Fig. 2d). Atg41 inactivation caused a decrease in transcription of *ATG8* but not *ATG1*, *ATG9* or *MET3*, a representative sulfur amino acid biosynthesis gene. Therefore, Atg41 seems to be involved in *ATG8* transcriptional induction, but our results do not support a general role in the transcriptional induction of *ATG* genes under sulfur starvation.

**Fig. 1 | Sulfur starvation induces autophagy. a** GFP-Atg8 processing assay. WT cells expressing GFP-Atg8 from *ATG8* endogenous promoter (Y1408) were grown to exponential phase in sulfur free (SF)-medium supplemented with 0.1 mM Met and starved as described in *Methods*. Cells were collected before (t0), and 2, 4, 6 and 8 h after starvation. Protein extracts were resolved by SDS-polyacrylamide gel electrophoresis and analyzed by western blot using antibodies against GFP and Pgk1 (loading control). Molecular weight markers are in kDa. Quantification was performed as described in *Methods*. GFP-Atg8 expression and GFP release are relative to the WT at 8 h. Data are mean of two independent cultures. **b** Live-cell microscopy. The strain and the starvation conditions are the same as above. Representative images are shown. The graphs indicate the percentage of cells showing GFP dots (left) and accumulating GFP fluorescence in the vacuole (right) Data are mean from two independent experiments using each time two different mutant clones, with in total 200–400 cells scored/ time point. Scale bar, 5 μm.

**c** Cell viability assay. Indicated strains (BY4742, Y1397, Y1424, Y1425, Y1426 & Y1406) were grown in SF-medium supplemented with 0.1 mM Met before sulfur starvation. Viability was determined as described in *Methods*. Data are mean ± SD of n independent cultures, n = 4 in the case of WT and *atg8Δ*, and 3 in the case of *atg1Δ*, *atg3Δ*, *atg7Δ*, and *atg9Δ*. **d** RNA-sequencing analysis. Cells (BY4742) were grown overnight in SF-medium supplemented with 0.1 mM Met and starved as described in *Methods*. Cells were collected by centrifugation before (t0) and 40, 80 and 120 min after the shift. RNA was extracted and processed for RNA-sequencing as described in *Methods*. The upper graph indicates maximum log2-fold change (FC) values (plain blue circle) and mean normalized counts (gray bars) calculated by the DESeq2 package for the 36 *ATG* genes. The lower graphs represent log2-FC values and normalized counts at the different time points for the top induced *ATG* genes (log2-FC > 3). Data are mean ± SD (log2-FC) or ± SEM (normalized counts), n = 3 RNA preparations from independent cultures. Source data are provided with this paper.

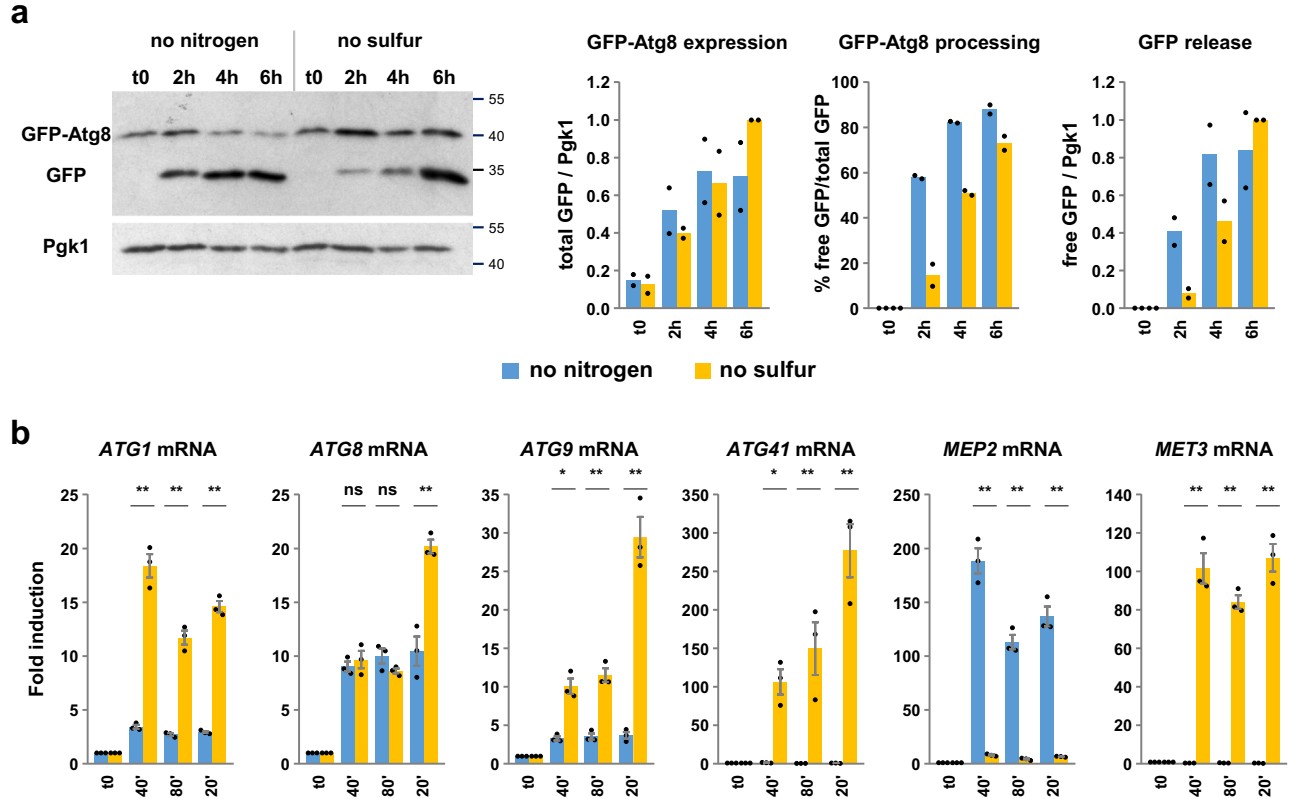

**Fig. 2 | Autophagy induction upon sulfur and nitrogen starvation involves distinct mechanisms. a** GFP-Atg8 processing assay. A prototrophic strain expressing GFP-Atg8 (Y1727) was grown into SF-medium supplemented with 0.1 mM Met and transferred into SF-medium without sulfur supplementation or into a derived medium lacking ammonium and amino acids except for 0.1 mM Met. Cells were collected before (t0), and 2, 4, and 6 h after starvation. GFP-Atg8 expression and GFP release are relative to the 6-h time point in no sulfur. Data are mean of *n* = 2 independent experiments. **b** Transcription analysis. A prototrophic

strain (Y1725) was subjected to starvation as above and transcription levels were measured by RT-qPCR. *MET3* and *MEP2* were used as specific control genes responsive to sulfur and nitrogen starvation, respectively. Fold induction is relative to t0. Data are mean ± SEM from n = 3 independent experiments. Statistical significance between conditions was determined by multiple two-sided *t* test comparisons using Holm-Sidak method. P values: **p < 0.0001; *p = 0.00037 and 0.00086; ns = 0.51 and 0.17 (from left to right). Source data are provided with this paper.

## Autophagy induction upon sulfur and nitrogen starvation involves distinct mechanisms

To investigate whether sulfur depletion-induced autophagy had distinctive characteristics, we analyzed autophagy flux and *ATG* gene transcription in cells exposed in parallel to sulfur or nitrogen starvation (Fig. 2). We used here prototrophic strains to exclude interferences from auxotrophic mutations. While GFP-Atg8 induction was similar between both conditions, processing was significantly more efficient under nitrogen depletion compared to sulfur depletion, especially in the first 2 hours (Fig. 2a). Transcription analysis showed more striking differences (Fig. 2b): following nitrogen depletion, only

*ATG8* was substantially induced, whereas following sulfur depletion, *ATG1*, *ATG8*, *ATG9* and *ATG41* were all strongly induced. Therefore, global induction of *ATG* gene transcription appears to be a distinctive characteristic of sulfur depletion not observed with nitrogen depletion.

## Met4 is essential to induce autophagy in response to sulfur starvation

Using a candidate approach, we next looked for the transcription factor(s) responsible for *ATG* gene induction upon sulfur depletion. Gcn4 is the only transcription activator that was involved in induction

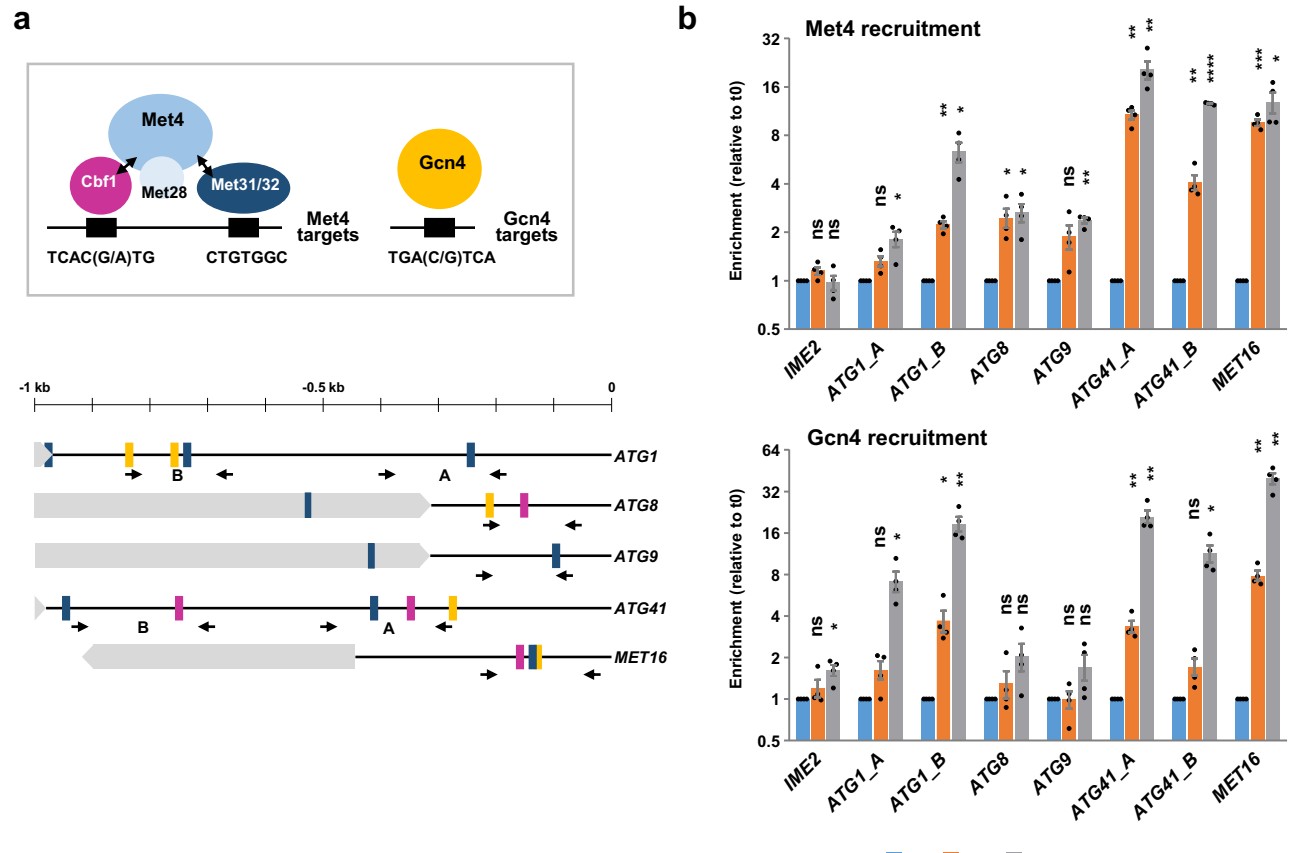

**Fig. 3 | The transcriptional activators Met4 and Gcn4 bind *ATG* genes in response to sulfur depletion. a** Upper drawing: schematics depicting the main mechanism of recruitment of Met4 and Gcn4 to their target promoters. Black boxes represent DNA-binding consensus sequences sites. Lower drawing: colored boxes indicate the positions of putative DNA-binding sites for Cbf1 (purple), Met31/Met32 (dark blue) and Gcn4 (orange) upstream of the initiation codon (0) of the indicated genes. Light gray boxes represent ORFs. Black arrows represent the primers used in the ChIP experiment. **b** Met4 and Gcn4 association with promoters. Cells expressing Myc-tagged Met4 or Gcn4 (Y1526 & Y1578) were grown and starved as in Fig. 1. Association with the indicated genes was measured by ChIP with antibodies against the Myc-tag. DNA was quantified by qPCR using primers specific for the promoter regions of *ATG1*, *ATG8*, *ATG9*, *ATG41*, and *MET16* and for the middle region of *IME2 ORF*. *MET16* and *IME2* were used as positive and negative controls, respectively. Data are mean ± SEM ($n = 4$). Statistical significance compared with t0 was determined by two-way ANOVA with Geisser-Greenhouse correction followed by Dunnett's multiple comparisons test. *P* values for Met4: ****$p < 0.0001$; ***$p = 0.0005$; **$p = 0.0026$, 0.0013, 0.0013, 0.0085, and 0.0096; *$p = 0.0426$, 0.0146, 0.0363, 0.0278 and 0.0136; ns = 0.16, 0.95, 0.07 and 0.11 (from left to right). For Gcn4: **$p = 0.0076$, 0.0093, 0.0048, 0.0029 and 0.0029; *p = 0.0417, 0.0234, 0.043, 0.0126; ns =0.481, 0.133, 0.556, 0.169, 0.999, 0.222 and 0.094. Source data are provided with this paper.

of certain *ATG* genes in response to amino acid starvation in *S. cerevisiae*[41]. On the other hand, the only transcriptional activator known to respond to sulfur availability in *S. cerevisiae* is Met4[42]. Whereas Gcn4 can bind its target promoters on its own, Met4 recruitment requires the basic helix-loop-helix factor Cbf1 and the two homologous zinc-finger factors Met31 and Met32[16,17]. Sequences matching the consensus binding sites for Gcn4, Cbf1 and Met31/32 are present upstream of several *ATG* genes, including *ATG1*, *ATG8*, *ATG9* and *ATG41* (Fig. 3a and Supplementary Table 7). To determine whether Gcn4 or Met4 bind these genes, we performed chromatin immunoprecipitation (ChIP) experiments using strains expressing C-terminal Myc-tagged proteins. In the case of Met4 (Fig. 3b, upper graph), we measured a strong enrichment of *ATG1* and *ATG41* promoter regions following sulfur depletion (by as much as 6-fold and 20-fold). We also observed a 2- to 3-fold enrichment of *ATG8* and *ATG9*, and no significant enrichment with the negative control *IME2*. In the case of Gcn4 (Fig. 3b, lower graph), enrichment of *ATG1* and *ATG41* promoter regions also increased quite strongly following depletion (by 19- and 21-fold, respectively). By contrast, there was no significant enrichment of *ATG8* and *ATG9*. These results strongly support that Met4 and Gcn4 bind *ATG1* and *ATG41* promoters upon sulfur starvation and suggest association of Met4 with *ATG8* and *ATG9*.

To determine whether Met4 and Gcn4 are required for transcriptional induction of *ATG1*, *ATG8*, *ATG9* and *ATG41*, we performed RT-qPCR in *met4* and *gcn4* single and double mutants subjected to sulfur starvation (Fig. 4a). Inactivation of Met4 or Gcn4 separately altered the transcription profiles of the four *ATG* genes; however, Met4 inactivation had a stronger effect compared to Gcn4 inactivation, especially in the case of *ATG9* and *ATG41*. The strongest effect was observed in the case of the double *met4Δ gcn4Δ* mutant, in which maximal induction of *ATG1*, *ATG8*, *ATG9* and *ATG41* were 5-, 3-, 6- and 9-fold lower, respectively, compared to the WT strain. Together with the ChIP data, these results support the hypothesis that Met4 and Gcn4 are both involved in the transcriptional induction of the four *ATG* genes under sulfur starvation, with Met4 having a preponderant role. The residual transcription in the double mutant may suggest the existence of additional players.

We next used the GFP-Atg8 and the Pho8Δ60 assays to assess the consequences of the transcriptional defects observed in the *met4Δ* and *gcn4Δ* mutants on the overall autophagic process. Gcn4 inactivation

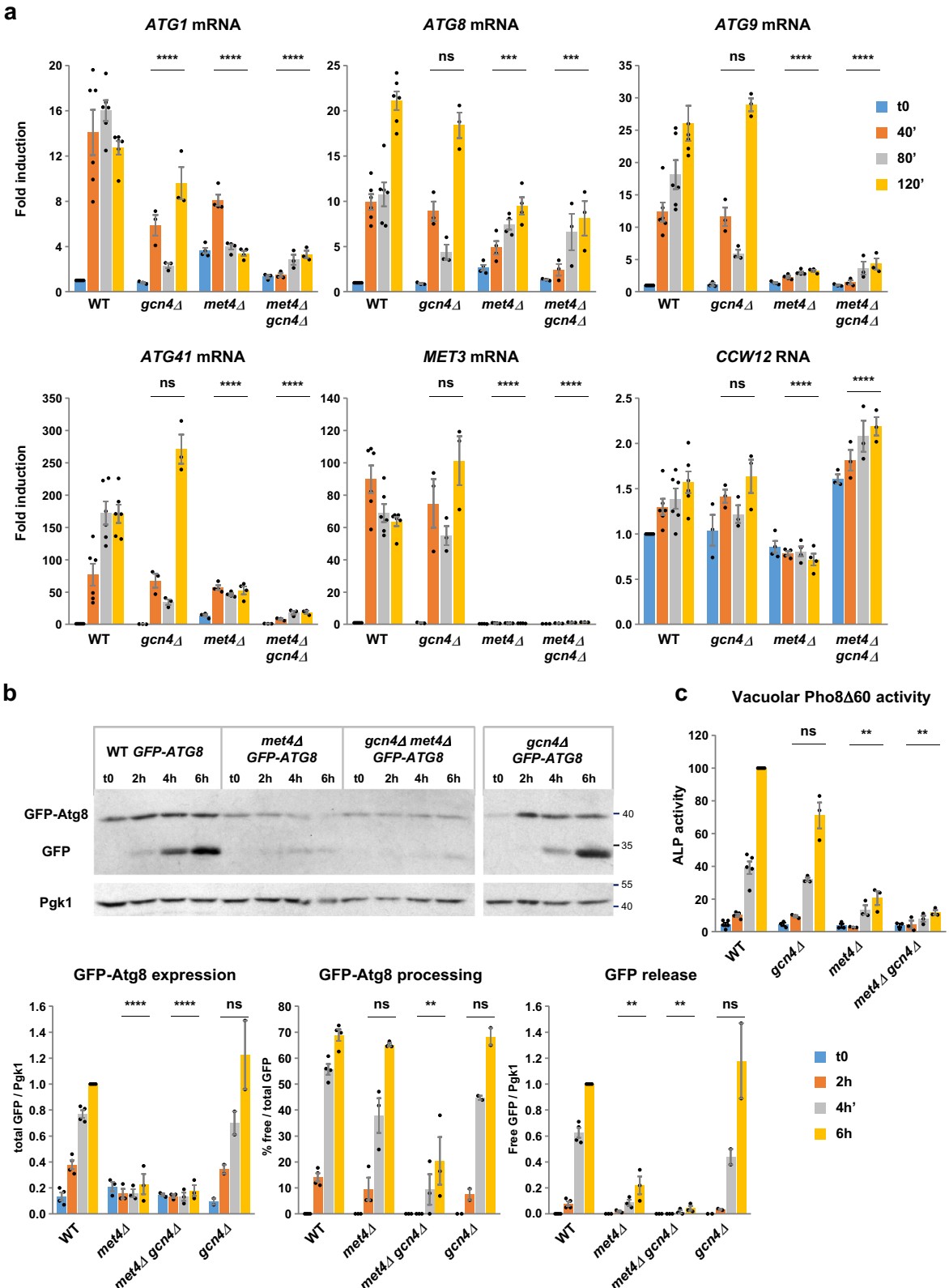

had no significant effect on neither expression nor processing of GFP-Atg8 (Fig. 4b) and caused only a slight decrease in ALP activity after 6 h of starvation (Fig. 4c). In contrast, Met4 inactivation caused a notable 5-fold decrease in GFP-Atg8 expression as well as free GFP levels (Fig. 4b), suggesting reduced autophagic activity, as further confirmed by the ALP activity measures shown in Fig. 4c. However, the persistence of low amount of free GFP indicated functional autophagy flux.

In the *gcn4Δ met4Δ* double mutant, GFP-Atg8 expression was as low as in the *met4Δ* single mutant, free GFP levels and ALP activity were further reduced, and GFP-Atg8 processing was significantly diminished (Fig. 4b, c). Similar results were observed in a strain carrying the *gcn4Δ* and *met4Δ* mutations in a prototrophic background (Supplementary Fig. 3). We also exposed the *met4Δ* and *gcn4Δ* prototrophic strains to nitrogen depletion and assessed autophagy flux using the GFP-Atg8

**Fig. 4 | Met4 is essential to induce autophagy in response to sulfur starvation.** **a** Transcription analysis. Indicated strains (BY4742, Y1437, Y1439 & Y1571) were grown to exponential phase in SF-medium supplemented with 0.1 mM Met and 0.01 mM SAM and starved as in Fig. 1. Transcription was measured by RT-qPCR on samples collected at the indicated times. *CCW12* was used as positive control. Fold induction is relative to WT at t0. Data are mean ± SEM (WT, *n* = 6; *met4Δ*, *n* = 4; *gcn4Δ* and *gcn4Δ met4Δ*, *n* = 3; n are independent experiments). Statistical significance compared with WT was determined by two-way ANOVA with Geisser-Greenhouse correction followed by Dunnett's multiple comparisons test. *P* values: ****$p$ < 0.0001; ***$p$ = 0.0006 and 0.0004; ns = 0.45, 0.76, 0.97, 0.99 and 0.99. **b** GFP-Atg8 processing assay. Indicated strains expressing GFP-Atg8 from *ATG8* endogenous promoter (Y1408, Y1611, Y1583 & Y1608) were grown and starved as above. Cells were collected at the indicated times and processed as described in

*Methods.* Molecular weight are in kDa. GFP-Atg8 expression and GFP release are relative to the value in the WT at the 6-h time point. Data are mean ± SEM (*n* = 3 independent experiments). Statistical significance compared with WT was determined as above. *P*-values, in each case from left to right: ****$p$ < 0.0001; **$p$ = 0.0012, 0.0058 and 0.0016; ns = 0.99, 0.79, 0.96 and 0.99. **c** Pho8Δ60 assay. Indicated strains expressing Pho8Δ60 under the control of *ADH1* promoter (Y1628, Y1718 or 1720, 1722 or 1765 & Y1723) were grown and subjected to sulfur starvation as above. Vacuolar Pho8Δ60 alkaline phosphatase (ALP) activity was measured as described in *Methods* and is relative to the WT at the 6-h time point. Data are mean ± SEM (WT, *n* = 5 and *gcn4Δ, met4Δ, gcn4Δ met4Δ, n* = 3). Statistical significance compared with WT was determined as above. *P* values: **$p$ = 0.0070 and 0.0029; ns = 0.73. Source data are provided with this paper.

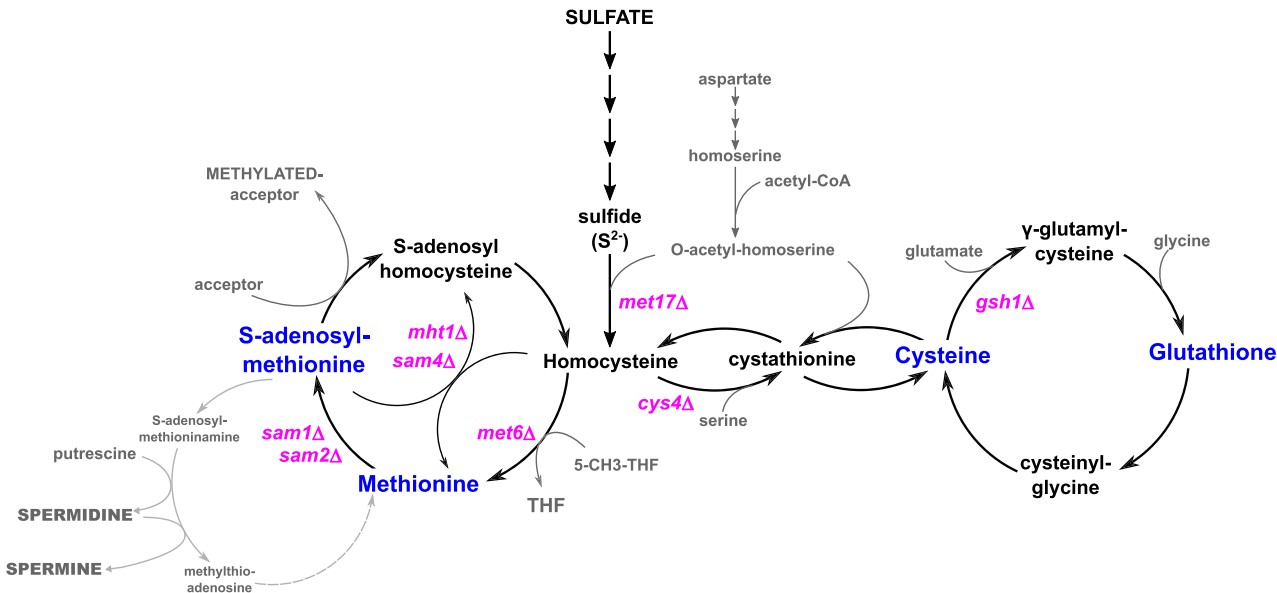

**Fig. 5 | Schematics of the sulfur-containing amino acid biosynthesis pathway showing the metabolic mutants used in this study.** The *met17Δ* mutation prevents assimilation of sulfate. Combination of *met6Δ*, *mht1Δ* and *sam4Δ* prevents methylation of homocysteine into methionine. The sam1Δ and sam2Δ mutations inactivate the two genes encoding methionine-adenosyltransferase (also known as

SAM synthetase), preventing SAM biosynthesis from methionine. The *cys4Δ* mutation inactivates cystathionine-β-synthase and blocks the transsulfuration pathway ensuring conversion of homocysteine (Hcy) into cysteine. The *gsh1Δ* mutation prevents GSH biosynthesis.

assay (Supplementary Fig. 4). We observed no significant effect of *met4Δ* and *gcn4Δ* mutations on GFP-Atg8 expression and GFP-Atg8 processing compared to the WT strain, suggesting that Met4 role in autophagy regulation under starvation does not extend to other nutrients besides sulfur. Finally, we measured the survival capacity under sulfur starvation of *met4Δ*, *atg1Δ* and *met4Δ atg1Δ* cells (Supplementary Fig. 3c). The three mutants showed a severe loss of viability in the 8 days following starvation compared to the WT strain. Strikingly, the amounts of *atg1Δ met4Δ* cells did not increase during the first day of starvation, contrary to the other strains. Moreover, the combined *atg1Δ met4Δ* mutations caused a significantly stronger decrease in survival than the simple *atg1Δ* mutation, indicating negative epistasis.

Altogether, our results show that Met4 plays a critical role in the autophagic process under sulfur starvation, whereas Gcn4 participates but is dispensable. They also support the notion that the cell capacity to overcome sulfur depletion involves interactions between Met4 and autophagy.

## Transcription of *ATG* genes is induced by depletion of organic sulfur

Depletion of sulfur from the growth medium causes deficiency of several key sulfur-containing metabolites, including methionine,

cysteine, SAM and GSH. Therefore, we investigated which metabolite was involved in signaling the induction of *ATG* genes. To test whether inorganic sulfate has a role by itself, we used the *met17Δ* mutant, which is blocked in the last step of the sulfur assimilation pathway and is therefore unable to synthesize organic sulfur compounds from sulfate (Fig. 5). Shifting WT cells from a medium containing methionine as sulfur source to a medium containing sulfate caused only moderate and transient activation of *ATG1*, *ATG8*, *ATG9* and *ATG41* (Supplementary Fig. 5). In contrast, transfer of *met17Δ* cells from methionine to sulfate caused a strong and sustained transcriptional activation of the four *ATG* genes, indicating that *ATG* genes respond to the depletion of organic sulfur under sulfur starvation.

## Cysteine or methionine depletion is not a major determinant of *ATG* gene induction

To achieve selective depletion of individual organic sulfur compounds, we used metabolic mutants (Fig. 5) placed in SF-medium supplemented with chosen sulfur sources. Inactivation of *GSH1* blocks the first step of GSH biosynthesis. The *gsh1Δ* cells were grown prior to depletion in a medium containing in addition to methionine GSH at very low concentration (5 μM) to sustain normal growth while avoiding GSH hyperaccumulation. RT-qPCR analyses showed that none of the *ATG* genes were activated in the 2-h period following the transfer in the

medium with no GSH, whereas strong activation occurred when all sulfur was depleted (Supplementary Fig. 6). Therefore, GSH depletion alone does not induce transcription of the *ATG* genes.

Inactivation of *CYS4* blocks the transsulfuration pathway responsible for methionine conversion into cysteine. Transfer of *cys4Δ* cells into a medium containing only methionine as sulfur source leads to cysteine depletion and rapid growth arrest (Supplementary Fig. 7b). Compared to the medium with no sulfur, transcription of the four *ATG* genes was induced with a delay and at significantly lower levels in this medium (Fig. 6a), indicating that cysteine depletion had only a limited effect on *ATG* gene expression. Western blot and fluorescence microscopy analyses of *cys4Δ* cells expressing GFP-Atg8 showed that cysteine depletion could induce autophagy flux, however GFP-Atg8 expression and free GFP levels were significantly lower in this condition compared to total sulfur depletion (Fig. 6b, c).

To block methionine biosynthesis, we deleted *MET6* together with *MHT1* and *SAM4*. *MET6* encodes methionine synthase, the main enzyme that catalyzes the conversion of homocysteine to methionine, using 5-methyltetrahydrofolate as methyl donor. The products of *MHT1* and SAM4 can also methylate homocysteine to give methionine but using SAM as methyl donor[43,44]. The *met6Δ mht1Δ sam4Δ* cells arrested growth similarly upon transfer in the medium deprived of methionine and the medium deprived of all sulfur supplements (Supplementary Fig. 7c). However, we observed a marked difference in *ATG1*, *ATG8*, *ATG9* and *ATG41* transcription between the two starvation conditions (Fig. 6d). *ATG1*, *ATG8* and *ATG41* very poorly induced and *ATG9* was not, under methionine starvation, whereas the four genes were strongly induced under total sulfur starvation. In line with these results, GFP-Atg8 expression and free GFP levels in the *met6Δ mht1Δ sam4Δ* mutant were significantly lower under methionine starvation than in the complete absence of sulfur, even though GFP-Atg8 was processed with similar efficiency in the two starvation conditions (Fig. 6e). At individual cell level, we observed an increase in the number of GFP dots as well as vacuolar accumulation of GFP under methionine starvation, but fluorescence signals were lower compared to sulfur starvation (Fig. 6f).

We further measured the effect of cysteine and methionine depletion on nonselective bulk autophagy specifically by using the 3-phosphoglycerate kinase (Pgk1)-GFP processing assay[45]. As expected, sulfur depletion in the WT cells induced autophagy-dependent processing of Pgk1-GFP (Supplementary Fig. 8a). Depletion of cysteine in the *cys4Δ* cells induced similar levels of Pgk1-GFP processing as depletion of total sulfur (Supplementary Fig. 8b). By contrast, depletion of methionine in the *met6Δ mht1Δ sam4Δ* cells induced very weak levels of Pgk1-GFP processing, 5 times lower compared to the depletion of total sulfur (Supplementary Fig. 8c). Curiously, the *met6Δ mht1Δ sam4Δ* mutant showed in the latter condition a delay in Pgk1-GFP processing and slightly lower percentage of free GFP compared to the other strains.

Altogether, our results indicate that individual depletion of the two sulfur amino acids fails to induce full transcription of *ATG* genes and triggers an incomplete autophagic response.

### SAM depletion strongly activates transcription of *ATG* genes in a Met4-dependent manner

To deplete SAM, we constructed a *sam1Δ sam2Δ* double mutant devoid of the two SAM synthetases found in *S. cerevisiae*. This mutant stopped growing rapidly in the medium which lacked SAM but still contained cysteine and methionine (Supplementary Fig. 7d). *ATG1*, *ATG8* and *ATG9* transcription was in this latter growth condition induced as much as in the total absence of sulfur (Fig. 7a). *ATG41* transcription was also strongly induced, albeit at a lower level. Therefore, SAM depletion was sufficient to trigger high transcriptional induction of the four *ATG* genes. Moreover, the *sam1Δ sam2Δ* cells subjected to SAM starvation showed levels of GFP-Atg8 expression, free GFP release and GFP-Atg8

processing comparable to the cells subjected to total sulfur starvation (Fig. 7b). We also observed similar increases between the two starvation conditions in the proportion of cells with GFP foci and in the proportion of cells accumulating GFP in the vacuole (Fig. 7c). Finally, Pgk1-GFP was processed almost as efficiently in the absence of SAM as in the complete absence of sulfur (Supplementary Fig. 8d). Therefore, SAM depletion, in addition to activate the *ATG* genes, also induces strong autophagic activity, even in conditions where the two sulfur amino acids are not limiting.

To establish whether Met4 was required upon SAM depletion, we carried out RT-qPCR in a *sam1Δ sam2Δ met4Δ* triple mutant. The *sam1Δ sam2Δ* double mutant and the WT strain were also included in the experiment for comparison. Met4 inactivation had a strong effect on the transcription of the four *ATG* genes upon transfer to the medium containing no SAM (Fig. 8). *ATG1*, *ATG8*, *ATG9*, and *ATG41* induction levels were on average 4, 7, 9 and 30 times lower, respectively, in the *sam1Δ sam2Δ met4Δ* cells than in the *sam1Δ sam2Δ* cells. We conclude that transcription induction of the *ATG* genes in response to SAM depletion depends strongly on Met4.

### SAM is sufficient to ensure full repression of *ATG* genes

To establish further the essential contribution of SAM in the transcriptional regulation of *ATG* genes, we performed depletion-repletion experiments with three combinations of mutations blocking different parts of the sulfur amino acid biosynthesis pathway: SAM synthesis, methionine synthesis, and both cysteine and methionine synthesis (Fig. 9). Cells were first starved for 90 min in SF-medium to induce autophagy and then cysteine, methionine and SAM were individually added. Repletion of the *sam1Δ sam2Δ* cells with methionine had no effect on the transcription levels of the four *ATG* genes, whereas repletion with SAM caused a rapid drop so that transcription levels were back to pre-starvation levels after 40 min (Fig. 9a). This drop in transcription levels could be due to SAM but also cysteine given that SAM conversion into cysteine is still possible in *sam1Δ sam2Δ*. However, addition of cysteine to the triple mutant *met6Δ mht1Δ sam4Δ*, which can transform cysteine into homocysteine but not into methionine and SAM, had only a limited effect on the transcription levels of the four *ATG* genes, far from the strong decrease obtained when adding methionine (Fig. 9b). Addition of cysteine to the quadruple mutant *met6Δ mht1Δ sam4Δ cys4Δ* had also a modest effect; by contrast addition of SAM to this mutant caused a strong decrease in the transcription levels of the four *ATG* genes (Fig. 9c), demonstrating that *ATG* gene repression by SAM does not require its conversion into cysteine or methionine. Altogether, these results emphasize the central role of SAM in the transcriptional regulation of autophagy in response to sulfur availability.

## Discussion

We show here that the regulation of autophagy in response to sulfur availability occurs in large part at the transcriptional level. Interestingly, comparison of cells deprived of either sulfur or nitrogen, revealed a clear difference in the way *ATG* genes respond in each case, in particular *ATG1* and *ATG9*, which are significantly more induced in the sulfur-deprived cells. This difference emphasizes the role of *ATG* gene transcriptional activation in the autophagic response to sulfur deprivation. By combining autophagy and transcription analyses upon depletion of individual sulfur compounds, we were able to assess the importance of *ATG* gene transcriptional induction in this particular autophagic response.

Individual depletions of methionine, cysteine and SAM are sufficient to trigger autophagy flux. However, the resulting autophagic activity, as assessed using the GFP-Atg8 and Pgk1-GFP cleavage assays, differ strongly between the three depletion conditions. Strikingly, methionine depletion has almost no effect on *ATG* gene transcription and autophagic activity, while SAM depletion causes strong

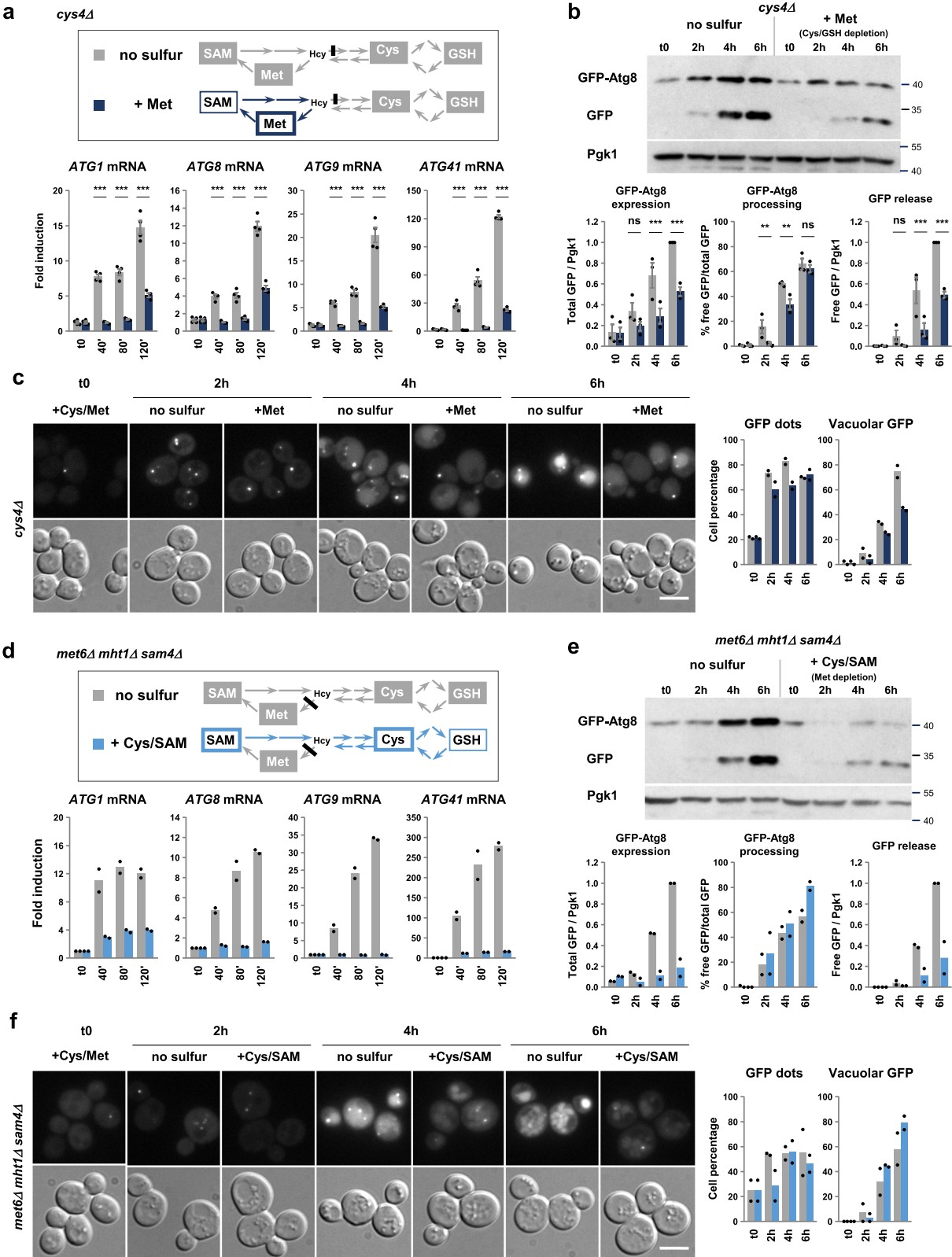

transcriptional induction and mimics alone the high levels of autophagic activity obtained upon depletion of all sulfur sources. By contrast, cysteine depletion has only a limited effect on *ATG* gene transcription but still induces some autophagic activity. Interestingly, levels of GFP-Atg8 expression and GFP release are lower upon cysteine depletion compared to sulfur depletion, while levels of cleaved Pgk1-GFP are similar. This suggests that cysteine depletion would

preferentially signal the induction of bulk autophagy whereas sulfur depletion would signal the induction of both bulk and selective autophagy, raising the possibility that the type of cargoes sequestered by autophagosomes might vary depending on the sulfur compound depleted. Moreover, our results as a whole suggest a model in which sulfur nutrients would regulate autophagy activity by affecting different steps. We propose that the two sulfur-containing amino acids

**Fig. 6 | Depletion of cysteine or methionine is not a major determinant of *ATG* gene induction.** Transcription analysis. Simplified schematics of the sulfur-containing amino acid biosynthesis pathway indicating the steps interrupted in the *cys4Δ* (**a**, upper panel) and *met6Δ mht1Δ sam4Δ* (**d**, upper panel) mutants. The sulfur compounds becoming depleted after transfer into the indicated starvation media are written in white on gray. Sulfur supplements are marked with a thick colored outline. The *cys4Δ* strain (Y1559 or 1677) was grown in SF-medium supplemented with 0.1 mM Met and 0.5 mM Cys, and shifted into SF-medium alone or containing 0.1 mM Met. The *met6Δ mht1Δ sam4Δ* strain (Y1533 or 1534) was grown in SF-medium supplemented with 0.1 mM Cys and Met, and shifted to SF-medium alone or containing 0.1 mM Cys and SAM. Samples were collected at the indicated times. Transcript levels (graphs) were measured by RT-qPCR. Fold induction is relative to t0. Data are mean of $n = 4$ in (**a**) or $n = 2$ in (**d**); n are independent experiments. Error bars indicate SEM. Statistical significance between conditions in (**a**) was determined by multiple two-sided *t* test comparisons using Holm-Sidak method. ***$p < 0.0001$.

**b, e** GFP-Atg8 processing assay. *cys4Δ* (Y1558) and *met6Δ mht1Δ sam4Δ* (Y1535) mutants expressing GFP-Atg8 from *ATG8* endogenous promoter were grown and shifted to starvation medium as above. Samples were collected at the indicated times. GFP-Atg8 expression and GFP release are relative to the 6-h time point in the no sulfur condition. Data are mean of $n = 3$ in (**b**) or $n = 2$ in (**e**); n are independent experiments. Error bars indicate SEM. Statistical significance between the two conditions was determined by multiple two-sided *t* test comparisons using Holm-Sidak method. *P* values: ***$p < 0.0001$; **$p = 0.00095$, 0.00017 and 0.00016; *$p = 0.0072$ and 0.0018; ns = 0.15, 0.43 and 0.25 (from left to right). **c, f** Live-cell imaging by fluorescence microscopy. The strains and starvation conditions are the same as above. Representative images are shown. The graphs indicate the percentage of cells showing GFP dots (left) and accumulating GFP fluorescence in the vacuole (right). Data are mean of $n = 2$ independent experiments using each time two different mutant clones, with in total 200–400 cells scored/ time point. Scale bar, 5 μm. Source data are provided with this paper.

would regulate autophagy mostly at a posttranscriptional step, for instance through protein modifications such as already described in the case of nitrogen starvation[35]. SAM would act above all at the transcriptional level, especially by regulating transcription of core *ATG* genes required for formation of autophagosome such as *ATG1*, *ATG8* and *ATG9*, resulting in a boost of autophagy levels.

Only a limited number of transcriptional regulators of *ATG* genes have been identified in *S. cerevisiae*, most of which are negative regulators[41,46–51]. We provide in this report evidence that Met4, the master transcriptional activator of *S. cerevisiae* sulfur amino acid metabolism, is directly involved in the regulation of autophagy. We show that Met4 binds and activates transcription of four *ATG* genes, including *ATG1*, *ATG8* and *ATG9*, which encode components fulfilling critical functions in autophagosome initiation. The fact that most *ATG* genes show an increase in transcription after sulfur depletion suggests that Met4 might bind and directly control more *ATG* genes than the four described in the present study. Using the GFP-Atg8 and Pho8Δ60 assays, we also show that Met4 inactivation does not block autophagy flux under sulfur starvation but decreases strongly the autophagic degradation activity, presumably because components of the autophagy machinery are still present and functional but in lesser amounts, which should result in smaller autophagosomes[52]. By contrast, Met4 does not seem to be required under nitrogen starvation conditions, indicating that its role does not extend to nutrient starvation in general. Met4 involvement in the regulation of both autophagy and sulfur metabolism may be seen as a way to allow coordination of the two processes, for example to sustain transformation of the sulfur amino acids generated by autophagy into other most needed sulfur compounds such as SAM and GSH.

How do cells sense and signal to the autophagy machinery that sulfur nutrients are insufficient? We provide evidence of a SAM-sensing and signaling pathway that connects sulfur nutrients availability to autophagy. First, we show that SAM depletion is sufficient to trigger induction of several key *ATG* genes even when sulfur amino acids are not limiting. Second, addition of external SAM to cells subjected to sulfur starvation rapidly stops transcription of the *ATG* genes even in genetic backgrounds where SAM cannot be transformed neither into methionine nor into cysteine. Importantly, SAM depletion does not only affect transcription of *ATG* genes but also leads to induction of the complete autophagic degradation process. Our results also provide the demonstration that SAM synthetase deficiency, which is in human the most common cause of persistent hypermethioninemia[53], can trigger autophagy induction.

Several types of mechanisms can be considered to explain how SAM regulates transcription of *ATG* genes. The positively charged sulfur atom of SAM is attached to three groups that can each be involved in enzymatic reactions: a methyl group, a 3-amino-3-carboxypropyl group and a 5'-deoxyadenosyl group[54]. The vast majority of reactions using SAM involves transfer of the methyl group[55]. The 3-amino-3-carboxypropyl group is required for synthesis of spermidine and spermine, and the 5'-deoxyadenosyl group serves as a source of radical intermediates in a number of biosynthetic reactions[56,57]. However, spermidine supplementation was shown to induce autophagy in several model systems including *S. cerevisiae*[58], indicating an antagonistic effect compared to SAM. Moreover, no radical SAM enzyme has been implicated in signaling so far[56]. Therefore, it is most probable that the transcriptional regulation of autophagy by SAM involves the methylation of some targets by a SAM-dependent methyltransferase. In particular conditions of induction of autophagy, Tu and colleagues[59] identified the carboxylmethyltransferase Ppm1 as a regulator of Protein Phosphatase 2 A (PP2A) in response to intracellular methionine/SAM levels. However, in our conditions, inactivation of Ppm1 has no effect on *ATG* gene activation and repression following sulfur depletion and then repletion (Supplementary Fig. 9), suggesting another type of mechanism. SAM could also be sensed by a protein possessing a SAM-binding domain but no enzymatic activity, like in the case of SAMTOR in mammals[60]. The SAM signal probably acts by interfering with Met4 since no *ATG* gene activation occurs in a mutant deficient for SAM synthetase and Met4 together. Possible mechanisms include downregulation of Met4 stability or downregulation of its nuclear localization, its recruitment to promoters or its activation capacity[61–64]. This could occur through direct methylation of Met4, or indirectly through methylation of the DNA-binding cofactors assisting Met4 recruitment or the ubiquitin-ligase SCF[Met30] known to downregulate Met4 activity and stability[61,62,65,66]. The SAM signal could also interfere with Met4 recruitment to *ATG* genes by affecting promoter accessibility, for example through changes in the methylation status of histones, as methylation levels of certain histones are linked to SAM levels[67]. Further investigation will be necessary to unveil the different actors involved in this SAM-sensing and signaling pathways.

## Methods

### Yeast strains and growth media

*S. cerevisiae* strains used in this study (Supplementary Table 1) derive from BY4742 and contain auxotrophic markers, unless otherwise stated in the text and figure legends. The individual deletion strains were obtained from the yeast knockout collection. Y1407 was generated by one-step integration at BY4741 chromosomal *ATG8* locus of *pP1KGFP-ATG8(406)*[52], which contains 990 base-pairs of *ATG8* promoter region in front of *GFP-ATG8* CDS. Other *GFP-ATG8-URA3* strains were constructed from Y1407 by crossing followed by diploid sporulation and tetrad dissection. Y1526 and Y1578 were obtained by genomic integration into Y1539 of PCR fragments amplified from Met4 and Gcn4 Myc-tagged strains previously published[68], followed by sporulation and tetrad dissection. Y1439 was obtained by sporulation and tetrad

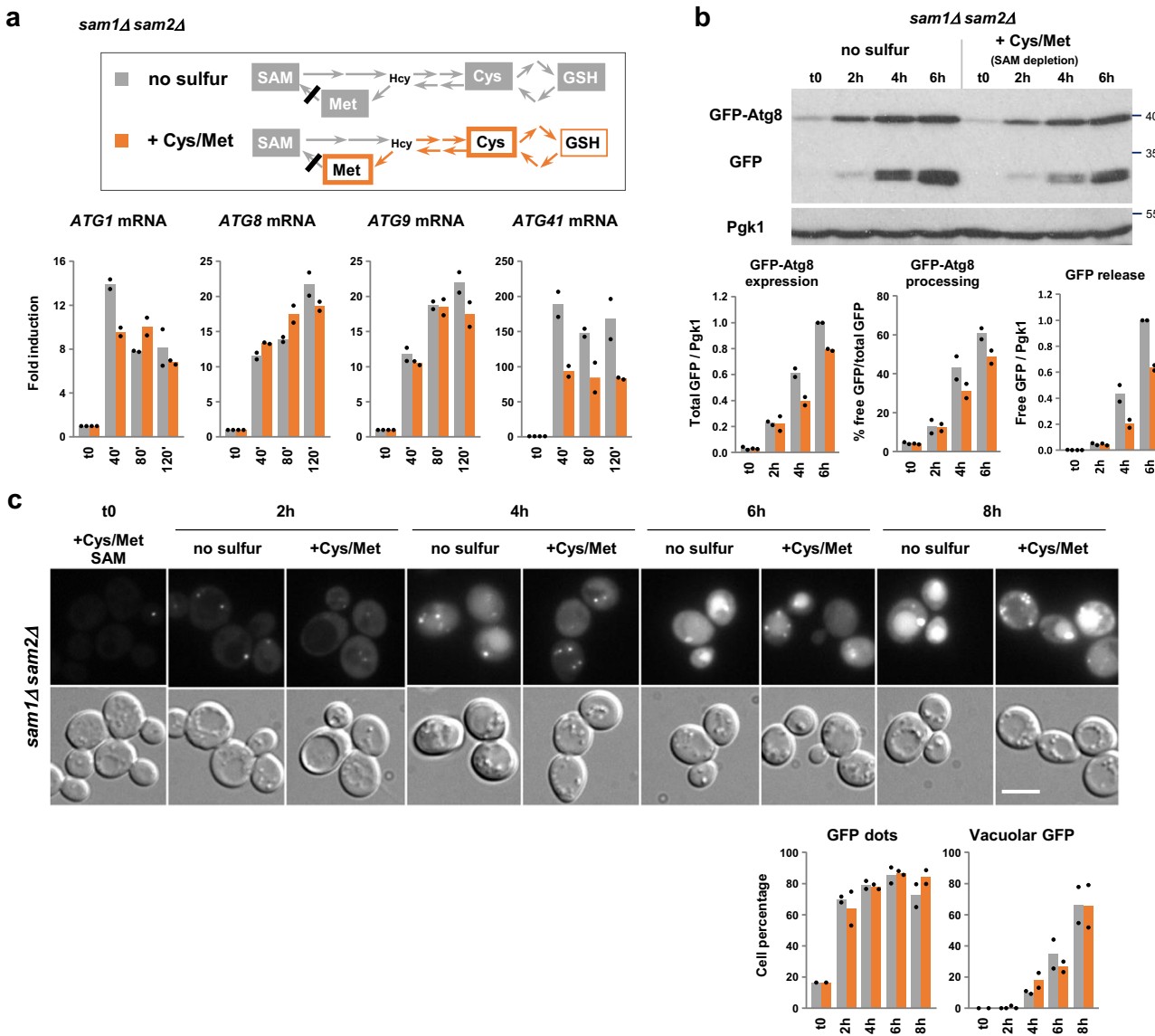

**Fig. 7 | SAM depletion fully activates transcription of *ATG* genes. a** Transcription analysis. Simplified schematics of the sulfur-containing amino acid biosynthesis pathway indicating the steps interrupted in the *sam1Δ sam2Δ* mutant (upper panel). The sulfur compounds that become depleted after transfer of the strain into the indicated starvation media are written in white on gray backgrounds. Sulfur supplements are marked with a thick outline. The *sam1Δ sam2Δ* (Y1504) mutant was grown in SF-medium supplemented with 0.1 mM Cys, Met and SAM, and shifted into SF-medium alone or containing 0.1 mM Cys and Met. Samples were collected at the indicated times. Transcript levels (graphs) were measured by RT-qPCR. Fold induction is relative to WT at t0. Data are mean of *n* = 2 independent experiments.

**b** GFP-Atg8 processing assay. The *sam1Δ sam2Δ* mutant expressing GFP-Atg8 from *ATG8* endogenous promoter (Y1506) was grown and starved as above. Cells were collected at the indicated times. GFP-Atg8 expression and free GFP are relative to the 6-h time point in the no sulfur condition. Data are mean of two independent cultures. **c** Live-cell imaging by fluorescence microscopy. The strain *sam1Δ sam2Δ GFP-ATG8* was grown and starved as above. Samples were observed at the indicated times. Representative images are shown. The graphs indicate the percentage of cells showing GFP dots (left) and accumulating GFP fluorescence in the vacuole (right). Data are mean of *n* = 2 independent experiments with in total 200-300 cells scored/ time point. Scale bar, 5 μm.

dissection on YPD medium supplemented with SAM of Y614. The double mutant *gcn4Δ::KanMX4 met4Δ::KanMX4* (Y1571) was obtained likewise using a diploid resulting from the crossing of two single mutants. Y1628 was obtained by genomic integration at the *PHO8* locus of BY4742 of a PCR fragment containing: *PHO8* upstream region, followed by *K. lactis URA3* gene, followed by *ADH1* promoter fused to a fragment of *PHO8* CDS starting at nucleotide position 181 (plasmid was kindly provided by Pr. Fulvio Reggiori). The mutant strains expressing Pho8Δ60 were derived by transformation with deletion cassettes and/ or crossing, sporulation and tetrad dissection. Y1504 and Y1506 were generated by two successive crosses. First, Y1490 and Y1491 were crossed with Y1407 to give diploids that produced, after sporulation and tetrad dissection, the haploids *MATα sam1Δ GFP-ATG8-URA3* and

*MATa sam2Δ*. Second, these two haploids were crossed and put to sporulate. Y1533 and Y1535 were constructed in three steps. First, Y1489 was crossed with Y1407 to produce, after sporulation of the diploid and tetrad dissection, a strain of genotype *MATa met6Δ GFP-ATG8-URA3*. In parallel Y1511 was transformed with a *mht1Δ::S.kluy.HIS3* PCR fragments amplified from CY51-1A[43] to create a strain of genotype *MATα mht1Δ::S.kluy.HIS3 sam4Δ::KanMX4*. Finally, the two strains were crossed and put to sporulate. Y1660 was derived from Y1533 by genomic integration of a *cys4Δ::LEU2* PCR fragment amplified from pUG73[69]. Y1681 was obtained by sporulation of a diploid resulting from the crossing of BY4741 with BY4742. The prototrophic strains Y1724 and Y1727 were obtained by sporulation of a diploid that was constructed by crossing Y1407 with Ura+ transformants obtained by

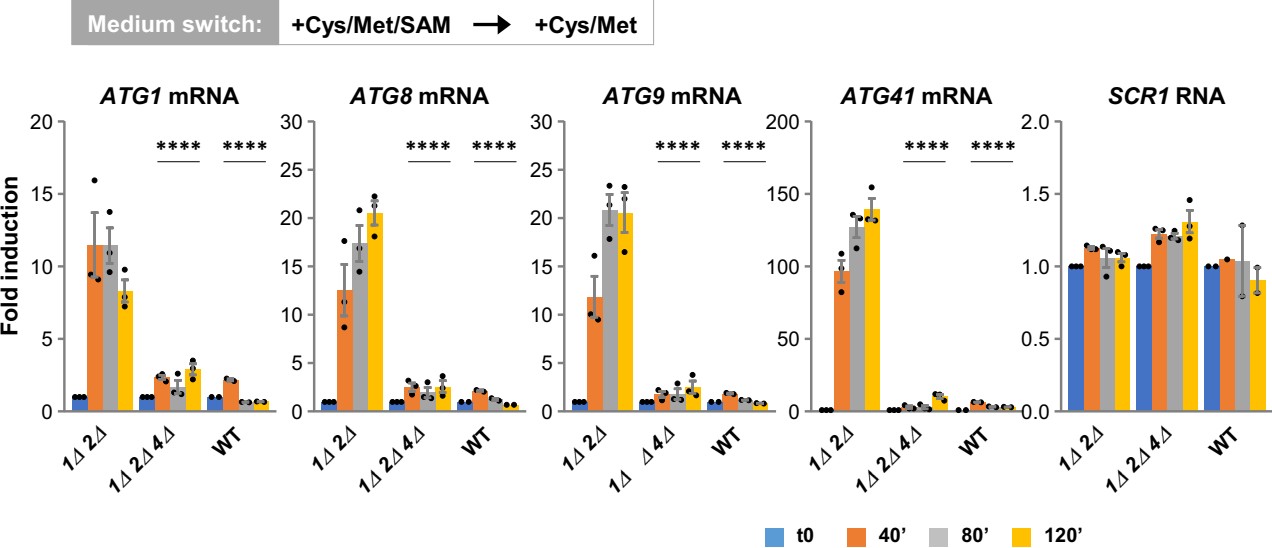

**Fig. 8 | Met4 has a key role in the transcriptional regulation of *ATG* genes by SAM.** *sam1Δ sam2Δ* (*1Δ 2Δ*), *sam1Δ sam2Δ met4Δ* (*1Δ 2Δ 4Δ*) and WT strains (Y1504, Y1601 & BY4742), were grown in SF-medium supplemented with 0.1 mM Cys, Met and SAM, and shifted to SF-medium supplemented with 0.1 mM Cys and Met. Samples were collected before (t0), and 40, 80 and 120 min after the shift.

Transcript levels (graphs) were measured by RT-qPCR. *SCR1* is used as positive control. Data are mean ± SEM of *n* = 3 independent experiments. Statistical significance compared with *sam1Δ sam2Δ* was determined by two-way ANOVA with Geisser-Greenhouse correction followed by Dunnett's multiple comparisons test. ****$p < 0.0001$. Source data are provided with this paper.

transformation of BY4720 (lys2Δ0 trp1Δ63 ura3Δ0) with a PCR fragment containing the wild-type *URA3* gene. The prototrophic deletion mutants containing *gcn4Δ*, *met4Δ* and *atg1Δ* null alleles were obtained by transformation with PCR fragments and/or crossing, sporulation and tetrad dissection.

Y1813, Y1831, Y1834 and Y1866 were constructed by one-step genomic integration in BY4742, Y1533, Y1677 and Y1504, respectively, of a *GFP-hphNT1* cassette amplified by PCR from pYM25[70]. Y1819 was constructed by one-step genomic integration in Y1813 of *atg1Δ::KanMX4* fragments amplified by PCR from Y1397.

YPD contained 1% yeast extract, 2% bacto-peptone and 2% glucose. YNB contained 0.7% yeast nitrogen base, 0.5% ammonium sulfate and 2% glucose. Solid medium also contained 2% agar. Sulfur-free (SF) medium was prepared from individual components based on YNB formula but replacing all sulfate salts by chloride salts (*see* Supplementary Table 2). YPD and YNB media were sterilized by autoclaving and SF medium by filtration through 0.2-μm-pore-size filters.

### Starvation protocol

YPD pre-cultures, inoculated with cells taken from agar plates freshly streaked from −80 °C stocks, were grown to late exponential phase during the day. A 1-mL aliquot was centrifuged, supernatant was eliminated by aspiration and cells were resuspended in 1 mL of fresh SF-medium. The suspension was then used to inoculate at low cell density 25 ml (120 mL for ChIP experiments) of SF-medium supplemented with the appropriate sulfur sources. Cultures were incubated overnight at 30 °C with orbital agitation (180 RPM; INFORS HT Multitron Standart) until OD$_{600}$ = 1–1.25 (around 10$^7$ cells/mL). To change medium, cultures were collected by filtration through Millipore 0.45-μm-pore-size HA-type membranes (except for viability assays; see below), SF-medium was passed through the membranes to wash the cells, and the membranes were transferred to SF-medium and incubation was resumed.

### Cell viability assay

Cells from 10-mL of culture were collected by 2-min centrifugation at 4500 RPM using a swinging bucket rotor, the supernatant was eliminated by aspiration, cells were then washed twice with 1 mL of SF-

medium and finally transferred into 10 mL of SF-medium. Cell viability, *i.e.* the ability to divide and proliferate[71], was assessed by quantification of colony forming units (CFU). Appropriate dilutions were spread out on YPD-plates (supplemented with SAM when necessary) with glass beads, and Colony-forming units (CFU) per mL were counted after 4 days of incubation at 30 °C.

### RNA isolation

Cells from 5 mL of culture at cell density 1–2 ×10$^7$/mL were collected by centrifugation and quickly frozen in liquid nitrogen. Cell pellets were kept at −80 °C until RNA extraction. Cells were resuspended in 400 μl of cold AE buffer (50 mM sodium acetate pH 5.3, 10 mM EDTA pH 8, 10% SDS) and the suspension was mixed with 400 μl of cold phenol saturated with 0.1 M citrate buffer pH 4.3 (Sigma). The mixture was incubated for 8 min at 65 °C with agitation, quickly frozen in liquid nitrogen, and incubated for another 8 min at 65 °C with agitation. Tubes were cooled down in ice and centrifuged at room temperature, 12000 rpm, for 10 min. The aqueous phase was then extracted once with 1 vol of acidic phenol:chloroform (1:1) and once with 1 vol of chloroform. Finally, RNA was precipitated with 0.1 vol of 3 M LiCl and 2.5 vol of absolute ethanol. After centrifugation at 12000 rpm and 4 °C for 10 min, RNA pellets were washed with absolute ethanol, air-dried and resuspended in autoclaved milli-Q water.

### RNA-sequencing

RNA quality was assessed on Agilent Bioanalyzer 2100 using RNA 6000 Pico kit (Agilent Technologies). 500 ng of RNA were treated with Baseline-ZERO DNAse (Epicenter) prior to ribosomal RNA depletion with the Ribo-Zero Yeast magnetic Kit (Illumina). Directional RNA-seq libraries were constructed using the Stranded Total RNA Preparation kit (Illumina). Libraries were pooled in equimolar proportions and sequenced (2 ×80 bp paired-end) on an Illumina NextSeq550 instrument using NextSeq 550 Mid Output kit.

Demultiplexing was done with the bcl2fastq2 conversion software (v2.18.12) and adapter sequences were removed with Cutadapt (v1.15). Only reads longer than 10pb were considered for further analysis. Between 12,890,040 and 17,482,126 reads were mapped to S288C genome (update R64.2.1) using the TopHat software (v2.1.1). Mapped

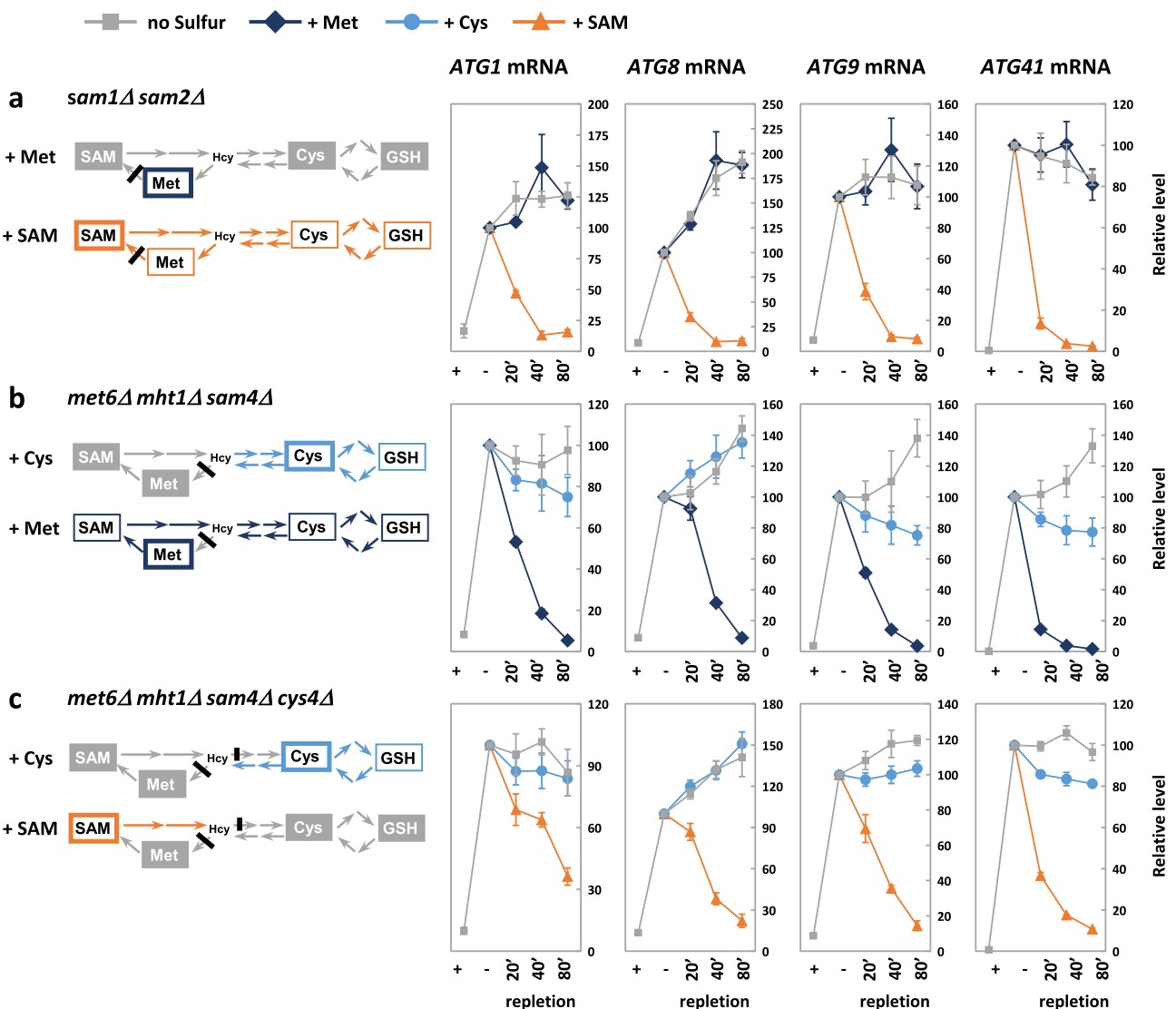

**Fig. 9 | Replenishment with SAM is sufficient to repress transcription of *ATG* genes in sulfur-starved cells.** The *sam1Δ sam2Δ* (**a**), *met6Δ mht1Δ sam4Δ* (**b**) and *met6Δ mht1Δ sam4Δ cys4Δ* (**c**) mutants (Y1504, Y1533 or 1534 & Y1660) were grown to exponential phase in SF-medium supplemented with 0.5 mM Cys, 0.1 mM Met and 0.1 mM SAM, and shifted to medium containing no sulfur. After 90 min, cells were replenished with 0.5 mM Cys, 0.1 mM Met or 0.1 mM SAM, as indicated. The schematics indicate: the sulfur component added after the 90-min starvation period (thick outline), the components that could be synthetized (thin outline),

and components that could not be synthetized due to the mutation and therefore remained depleted (written in white on gray backgrounds). The Samples were collected before the shift (+), following the starvation period (−) and at the indicated time points following repletion. Transcript levels (graphs) were measured by RT-qPCR, and are relative to the 90-min total starvation time point. Fold induction is relative to t0. Data are mean ± SD of *n* = 3 independent experiments. Source data are provided with this paper.

reads were counted using featuresCounts from the Subread package (v2.0.1). For each sample, between 75 and 80% of sequencing reads were successfully assigned. Differential analysis of read counts across time points was performed with the DESeq2 package[72], which uses the Walt test and the Benjamini-Hochberg procedure for hypothesis testing and adjusted *p* value (padj) calculation.

### RT-qPCR
Reverse Transcription (RT)-quantitative PCR was conducted following a two-step protocol. RevertAid reverse transcriptase (Thermo Scientific) was used, with random hexamers for priming, for the RT step, and Luna Universal qPCR Master Mix (Biolabs) and the LightCycler 480 instrument (Roche) for the qPCR step. Data were analyzed with the LightCycler 480 software by performing Absolute Quantification using an in-run standard curve and the Second Derivative Maximum analysis method. Sequence of primers is given in Supplementary Table 3.

### Chromatin immunoprecipitation (ChIP)
Cells in 40 mL of culture at 1–2 ×10⁷ cells/mL were fixed with 1% formaldehyde for 15 min at 30 °C. Fixation was stopped by addition of 0.4 M glycine, and cells were collected by centrifugation, washed with cold Tris-Hcl (20 mM, pH 8.0) and kept at -80 °C. Cell pellets were resuspended in 0.5 mL of FA-lysis buffer (50 mM Hepes-KOH, pH7.5, 150 mM NaCl, 1 mM EDTA, 1% Triton X-100, 0.1% deoxycholic acid sodium salt, 0.1% SDS) containing 1 mM Pefablock SC (Roche), and disrupted in the presence of 425-600 µm glass beads (Sigma-Aldrich) using a FastPrep FP120 instrument (Qbiogene). Lysates were recovered and centrifuged (20,100 g, 60 min, 4 °C) to pellets the crosslinked chromatin. Pellets were subjected to sonication in 1.8 mL of FA-lysis buffer using a 500 W Vibra-Cell ultrasonic processor (Sonics and Materials, Inc.) equipped with a 3-mm stepped microtip (8 cycles of 20 s at amplitude 30%, 4 °C), yielding DNA fragments ranging from 100 to 1000 base pairs with an average of 400 base pairs. The insoluble

debris was finally eliminated by centrifugation (13,400 g, 10 min, 4 °C), and chromatin extracts were kept at -80 °C.

Immunoprecipitation was performed by overnight incubation at 4 °C of 400 μL of chromatin extract with 2 μg of c-Myc mouse monoclonal antibody (clone 9E10, Santa Cruz Biotechnology, ref. sc-40). Immune complexes were collected with protein A-Sepharose (GE healthcare, ref. 17–0780.) and washed five times in FA buffer containing 0.5 M NaCl, followed by one time in TE (25 mM Tris-HCl pH 8.0, 5 mM EDTA). Immunoprecipitates were eluted from the protein A-Sepharose beads by incubating for 20 min at 65 °C in a solution containing 25 mM Tris-HCl pH7.5, 5 mM EDTA and 0.5% SDS. Eluates were decrosslinked overnight at 65 °C after addition of 1 mg/mL of Pronase (Roche). DNA was purified using GeneJET PCR purification kit (Thermo Scientific), and eluted in 100 μl of Tris-HCl (10 mM, pH 8.0).

DNA was quantified by real time PCR using the LightCycler 480 instrument (Roche) and Luna Universal qPCR Master Mix (Biolabs). Primer sequence is given in Supplementary Table 4. A typical run included 2 μl of immunoprecipitated (IP) and input (total) DNA in duplicate and serial dilutions of one input DNA to create a standard curve and determine PCR efficiency. Data were analyzed with the LightCycler 480 software by performing Absolute Quantification using the in-run standard curve and the Second Derivative Maximum analysis method. To calculate enrichment, we divided the qPCR quantification value for IP by the qPCR quantification value for the corresponding input.

### GFP-Atg8 processing assay

Cells from 5 mL of culture at $1-2 \times 10^7$ cells/mL were collected by centrifugation and quickly frozen in liquid nitrogen. Cell pellets were kept at -80 °C. Extract preparation and western blotting were carried out according to Guimaraes et al. [73]. Cells were broken by shaking in 200 μL of cold 10% trichloroacetic acid (TCA) in the presence of glass beads on a Vortex for 10 min at 4 °C, and the lysates were centrifuged at 13000 RPM for 5 min at 4 °C. Pellets were resuspended in 2X Laemmli sample buffer and heated for 5 min at 95 °C. A volume corresponding to $1.5 \times 10^7$ cells was separated on 12% SDS-polyacrylamide gel, transferred onto nitrocellulose membrane, and probed with antibodies against GFP (clones 7.1 and 13.1, Roche ref. 11814460001, dil. 1:1,000) and Pgk1 antibodies (clone 22C5D8, Invitrogen ref. 459250, dil. 1:1,000), followed by peroxidase-conjugated anti-mouse antibodies. Detection was performed with Amersham ECL Western Blotting System, and signals were captured on films. Scans were quantified using ImageJ.

### Pgk1-GFP processing assay

Cells from 5 mL of culture at $1-2 \times 10^7$ cells/mL were collected by centrifugation and quickly frozen in liquid nitrogen. Total cell protein extracts were prepared by TCA precipitation. Cells were resuspended in cold 10% TCA and broken in a Precellys Evolution (Bertin Technologies, France) bead beater equipped with a Cryolys cooling system (settings: $6 \times 15$ s at 6,500 RPM, with 30 s of pause between cycles, 4 °C). Cell extracts were recovered, centrifuged at 13,000 RPM for 10 min, 4 °C, and pellets were resuspended in 2X Laemmli sample buffer. Western blots were performed as described above using mouse monoclonal antibodies against GFP (clones 7.1 and 13.1, Roche ref. 11814460001, dil. 1:1,000) and GAPDH (clone 1E6D9, Proteintech ref. 60004-1.-Ig, dil. 1:10,000), goat anti-mouse IgG HRP-conjugated antibodies (Santa Cruz Biotechnology ref. sc-2005, dil. 1:20,000), SuperSignal West Pico PLUS detection reagents (Thermo Fisher Scientific), and the ChemiDoc Imaging System (Bio-Rad Laboratories). Bands were quantified by densitometric analysis using the ImageJ software.

### Pho8Δ60 assay

Cells from 5 mL of culture at $1-2 \times 10^7$ cells/mL were collected by centrifugation and quickly frozen in liquid nitrogen. Cell lysates were prepared, and alkaline phosphatase (ALP) activity measured, according to Araki et al. [74]. Briefly, cells were resuspended in ice-cold

assay buffer (250 mM Tris-HCl, pH 9; 10 mM MgSO4 and 10 μM ZnSO4) containing 1 mM PMSF and broken by vortexing in the presence of glass beads for 15 min at cold temperature. For the assay, 10 μl of the clarified lysate were added to 500 μl of ice-cold assay buffer and placed at 30 °C for 5 min before adding 50 μl of 55 mM α-naphthyl phosphate disodium salt (Merck). The reaction was stopped after 20 min with 500 μl of 2 M glycine-NaOH, pH 11 and the fluorescence measured at wavelength of 345 nm for excitation and 472 nm for emission using Tecan's Infinite 200 Pro plate reader. For normalization, the protein concentration of cell lysates was determined using the Bradford reagent from Sigma-Aldrich. The ALP activity corresponds to the emission per the amount of protein in the reaction (mg) and the reaction time (min).

### Fluorescent microscopy

Small (1-mL) cell aliquots were transferred into 1.5 mL tubes and left on the bench at 30 °C for 15 min before preparing the slides to allow cells to settle. Cells were observed using a three-dimensional deconvolution microscope (DMIRE2, Leica Microsystems) equipped with a 100x oil objective (HC PL APO 100x/1.4 oil CS, Leica Microsystems) and an incubation chamber set to 30 °C. Images were captured using a 20-MHz CoolSNAP HQ2 CDD camera (Roper Technologies) with a z-optical spacing of 0.2 μm. Z-series acquisition (21 images per stack) and raw image deconvolution was performed using Metamorph software (Molecular Devices). Images were processed using ImageJ.

### Statistical analyses

Graphs and statistics were performed using GraphPad Prism 8.0 software. The number of independent biological replicates (n) is indicated in the figure legends. Error bars indicate standard error of the mean (SEM) when individual values are shown and standard deviation (SD) when they are not. Statistical significance was assessed as detailed in the figure legends by two-way ANOVA followed by the recommended post hoc test or by multiple two-sided $t$ test comparisons. Statistical significance is given in figure legend.

### Reporting summary

Further information on research design is available in the Nature Portfolio Reporting Summary linked to this article.

## Data availability

Numerical data and Western blots are provided as a Source data file with this paper. RNA-sequencing data were deposited in GEO under accession code GSE204733. Source data are provided with this paper.

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

## Acknowledgements

We thank: Yongheng Liang (Nanjing Agricultural University) for plasmid *pP₁ₖGFP-ATG8(406)*; Fulvio Reggiori (Aarhus University, Denmark) for the plasmid containing the construct *PHO8*[-647,-156]-*K.lac.URA3-ADH1*[-720,-15]-*PHO8* [181-466]; Yan Jaszczyszyn and Kévin Gorrichon for RNA-seq libraries preparation and sequencing; Loic Quenechdu for his assistance in the construction of strain Y1533; Virginie Chiodelli and Fanny Culot for glassware and media preparation. We acknowledge the initial support of I2BC scientific council, and the sequencing and bioinformatics expertise of the I2BC Next Generation Sequencing facility. We are grateful to Svetlana Dokudovskaya (Institut Gustave Roussy, Villejuif), Nadine Camougrand (IBGC, Bordeaux) and Lisete Fernandes (BioSystems and Integrative Sciences Institute, Lisboa) for constructive comments and suggestions on the manuscript. Financial support was provided by I2BC, Center National de la Recherche Scientifique (CNRS) and Institut National de la Santé et de la Recherche Médicale (INSERM). I2BC Next Generation Sequencing facility is member of France Génomique funded by ANR-10-INBS-09.

## Author contributions

L.K. conceived the study with R.L. and S.C., supervised the project and wrote the manuscript with inputs from all other authors. L.K., M.P., S.C., M.-H.C., and R.L. designed the experiments and analyzed the results. M.P., H. J.-J., and L.K. constructed the strains and performed the experiments. D. N. helped with the design of the RNA-seq experiment and performed the bioinformatic analyses.

## Competing interests

The authors declare no competing interests.
