## [Peer Review File · Nature Communications]

Sulfur starvation-induced autophagy in *Saccharomyces cerevisiae* involves SAM-dependent signaling and transcription activator Met4REVIEWER COMMENTS

Reviewer #1 (Remarks to the Author):

The manuscript entitled « Sulfur starvation-induced autophagy in *Saccharomyces cerevisiae* involves SAM-dependent signaling and transcription activator Met4” by Prigent and co-workers reports the transcriptional induction of autophagy-genes in response to sulfur limitation. They show an important role for the transcription factor Met4 in this process and for the metabolite S-adenosyl methionine. Depletion of methionine or cysteine also had an additional post-transcriptional effect, indicating that sulfur limitation results both in an increase of the autophagy flow rate through post-transcriptional activation, but also an increase of the autophagy flux capacity through transcriptional activation of several ATG genes. The underlying molecular mechanisms remain to be elucidated. This is a nice and globally solid piece of work. The experiments are sound, the results are clear and support the conclusions. I have several concerns which are detailed below. Addressing experimentally these various points would, in my opinion, strengthen the conclusions.

Major points:

1- All the work is done using yeast strains carrying several auxotrophic markers (mutations in amino acids and nucleotide synthesis pathway genes) that could interfere with the observations. I understand that these “classical” lab strains are more convenient to use but it seems to me that some of the key experiments should be repeated in prototrophic yeast strains to ensure that the conclusions are robust and physiologically correct. This is of particular importance for Fig. 1a, Fig. 3c and supplemental Fig. 3.

2- Strains used in each experiment should be strictly isogenic. This is apparently not the case although difficult to figure out in several cases (see point 5 below). Using non isogenic strains may severely affect the conclusions and should be avoided. For example, in supplemental Fig. 3, the two strains BY4741 and BY4742 differ at the LYS2 locus in addition to the MET17 (MET15) locus which is the one tested in the experiment. The *lys2* auxotrophy strongly affects yeast physiology.

3- Although elucidating the molecular underlying mechanisms is not in the scope of this work, it would be nice to have some experiments done under sulfur and nitrogen starvation conditions in parallel, to establish which observations are specific to sulfur starvation. This is of particular importance for Fig. 1c, Fig. 3b, supplementary Fig. 2a. Specifically, the authors should address the following questions: do the responses to these two nutritional signals share some pathway components? For example, do the sulfur regulation signals need a functional TOR pathway? Is autophagy normally induced in a *met4* mutant in response to nitrogen starvation? A few simple experiments would allow to widen the conclusions and give a preview of what is taking place (or not) at the molecular level.

4- The role of Atg41 in the response to sulfur starvation is unclear. Data in supplementary Fig.2a show that the *atg41* mutation blocks GFP-Atg8 processing indicating a role of Atg41 in autophagy (as expected), but also blocks GFP-Atg8 expression suggesting that a functional autophagy pathway might be required for the transcriptional response to sulfur limitation to

occur. This should be clarified by investigating whether other Atg mutants (affecting different steps of the autophagy process) have the same effect and whether they affect expression of other Met4 target genes (MET genes).

Minor points:

5- The name of the strains used in each figure should be clearly specified. For example, in Fig. 1c, "WT" could refer to several strains in supplementary Table1.

Reviewer #2 (Remarks to the Author):

In the current manuscript, the authors analyze the autophagy response of budding yeast upon sulfur starvation. They identify a strong transcriptional response for autophagy genes mediated by the known central transcription factor Met4 in the absence of sulfur. Atg41, a factor highly upregulated upon sulfur starvation in a Met4-dependent manner, plays a critical role for Atg8 expression and autophagy flux. The authors convincingly dissect the possible metabolic inputs for autophagy regulation and identify S-adenosylmethionine as a key metabolite.

The authors describe an interesting dataset and use systematic yeast genetics to identify the key metabolite SAM for sulfur starvation-induced autophagy. However, the study falls short in providing initial mechanistic insights into how SAM levels might be connected to the regulation of Met4 and/or autophagy and, importantly, ignores published data that have linked SAM to autophagy regulation, as described below.

Critical points:

(1) SAM has been linked to the regulation of autophagy in a previous publication at mechanistic level; SAM is a substrate for the methyltransferase Ppm1, which regulates the protein phosphates PPA2; methylated PPA2 promotes dephosphorylation of Npr2, a negative regulator of TORC1 and positive regulator of autophagy (Sutter et al. Cell 2013). It is inappropriate that the authors cite this article, when speculating about a link of TORC1 and Met4 regulation, but do not mention the described role of SAM for autophagy regulation. Clearly, these mechanisms might also underly autophagy and/or Met4 regulation during sulfur starvation-induced autophagy.

(2) Overall, the manuscript lacks insight into how Atg8 protein levels, autophagy flux and the transcriptional program are linked and/or their mechanistic bases. How SAM levels are translated into Met4 and/or autophagy flux are key questions.

(3) The essential role for Atg41 for sulfur starvation-induced autophagy is really interesting. In particular, it is interesting that Atg8 levels and turnover are significantly impaired in the absence of Atg41. Is the effect on Atg8 protein levels specific to the absence of Atg41 or does a block in any core ATGs cause these effects?

Reviewer #3 (Remarks to the Author):

Summary

Prigent et al. provide evidence that the master regulator of sulfur metabolism in yeast, Met4, activates the transcription of ATG genes in response to sulfur starvation, and suggest that this signaling is specifically in response to SAM depletion. The authors also demonstrate the importance of sulfur starvation-induced autophagy for cellular viability after prolonged sulfur deprivation.

Validity

In general, the interpretation of data and conclusions drawn by the authors are reasonable. While the ChIP experiments demonstrate that Met4 is recruited to the ATG genes examined, it is less obvious that Met4 is affecting the mRNA levels of these genes. There appears to be significant alteration in the levels of ATG mRNA during the time course (e.g., up 40 min, down 80 min, up again 120 min) in Fig 3A for the WT strain that make data interpretation and significance difficult to assess, in particular with respect to the *met4* and *gcn4* single deletions. Can the authors explain this?

Significance

The work done by Prigent et al. potentially represent a significant advance in the understanding of the mechanisms governing transcriptional regulation of autophagy. However, some of the data are not compelling and could be supported by additional experiments. Perhaps the authors might be overlooking some alternative mechanisms.

Data and Methodology

The figures and data are organized and presented clearly. The utilization of specially formulated "sulfur free" media is a strength of their methodology, as well as the use of specific sulfur metabolism mutants to identify the responsible metabolite. However, one concern the authors should look into is the use of BY strain background that may be auxotrophic for *met15Δ*, which could alter the response to sulfur and methionine metabolism or synergize with *met4Δ* deletion. They should consider validating key results using a prototrophic strain or other strain background such as W303.

Analytical Approach

The authors should state the n for their RT-qPCR experiments, unless I have missed it.

Suggested Improvements

Major points:

The authors rely heavily on the GFP-Atg8 cleavage assay to monitor autophagy. While some results obtained with this assay are consistent with their model, the marked reduction in the levels of GFP-Atg8 protein in the *met4* and *gcn4* single and double mutants compared to WT makes it difficult to assess levels of autophagy by the very nature of the mutants they are testing. This also applies to their microscopy data and other data where the growth condition alters the abundance of GFP-Atg8. While this reduction in GFP-Atg8 is informative in and of itself, the use of a complementary and more quantitative method to monitor

autophagy such as the Pho8 Δ 60 alkaline phosphatase assay would strengthen the work. After all, the induction of ATG genes and protein amounts may not matter in the end if ATG8 is not processed, etc. and there is comparable delivery of autophagosomes to the vacuole. To this point, the extent of processing of ATG8 for various mutants does not appear that different, when it might be expected that methionine addition should repress actual autophagy (e.g., Fig 5b, 5e) – these observations would appear inconsistent with the authors initial finding that sulfur starvation induces autophagy.

The authors provide evidence that Met4 may bind several ATG genes using CHIP, however the prominent induction of ATG genes can still be observed in the absence of Met4. This is quite confusing and suggests that the observed Met4 binding may not be functionally relevant. If Met4 were indeed a regulator of ATG genes, one might expect effects on mRNA levels similar to the extent observed with the MET3 transcript. Can the authors also observe binding of Cbf1, Met31 or Met32? Although significant effects were observed in the double *gcn4 met4* knockout, there is the concern that this strain could be quite sick as *Gcn4* is generally required for transcriptional response to amino acid starvation.

Minor points:

The authors should consider including growth rates of various mutants under normal versus starvation conditions.

Grammar and spelling on lines 55 (called the autophagosome), 70 (diverse), 111 and 310 (let hypothesize)

Clarity and Context

The text was written with clarity and context.

References

The authors reference appropriate literature.

Responses to reviewers' comments on manuscript NCOMMS-22-17321

Reviewers' comments are in blue; responses are in black; modified figures are indicated in bold. Addition introduced in the manuscript itself are marked in dark red and rephrasing in green.

Reviewer #1:

The manuscript entitled « Sulfur starvation-induced autophagy in *Saccharomyces cerevisiae* involves SAM-dependent signaling and transcription activator Met4” by Prigent and co-workers reports the transcriptional induction of autophagy-genes in response to sulfur limitation. They show an important role for the transcription factor Met4 in this process and for the metabolite S-adenosyl methionine. Depletion of methionine or cysteine also had an additional post-transcriptional effect, indicating that sulfur limitation results both in an increase of the autophagy flow rate through post-transcriptional activation, but also an increase of the autophagy flux capacity through transcriptional activation of several ATG genes. The underlying molecular mechanisms remain to be elucidated. This is a nice and globally solid piece of work. The experiments are sound, the results are clear and support the conclusions. I have several concerns which are detailed below. Addressing experimentally these various points would, in my opinion, strengthen the conclusions.

We thank the reviewer for the positive assessment.

Major points:

1- All the work is done using yeast strains carrying several auxotrophic markers (mutations in amino acids and nucleotide synthesis pathway genes) that could interfere with the observations. I understand that these “classical” lab strains are more convenient to use but it seems to me that some of the key experiments should be repeated in prototrophic yeast strains to ensure that the conclusions are robust and physiologically correct. This is of particular importance for Fig. 1a, Fig. 3c and supplemental Fig. 3.

Response: Most published studies on autophagy in yeast used strains with auxotrophic mutations. It may not be a problem in the case of studies focusing on the core autophagic process, but it might in the case of those interested in regulatory aspects. A strain containing auxotrophic mutations will surely not sense nitrogen starvation in the same manner as a prototrophic strain. The former should be more sensitive to the amino acids that it cannot synthesize and trigger additional signaling pathways in their absence. We believe that it should be less of a concern in the case of sulfur depletion from a medium containing all amino acids, but the concern still exists considering how metabolic pathways are intertwined. Therefore, we have repeated several experiments in prototrophic strains derived from BY4742. We have performed autophagy flux (GFP-Atg8 assay), transcriptional and viability analyses following sulfur depletion (**Fig. 2** and **supplemental Fig. 3**). We

have obtained similar results as with BY4742, indicating that the auxotrophic mutations present in BY4742 do not affect the autophagic response to sulfur depletion. We have also repeated autophagy flux and transcriptional analyses in prototrophic strains containing *gcn4* and *met4*-null alleles (**supplemental Fig. 3**). The results of these experiments support our conclusion that Met4 is essential for *ATG* gene induction upon sulfur starvation. Viewing these new results, we have not considered that reproducing the experiments using the *met17Δ* mutation (supplementary fig. 3 in the initial manuscript) in a prototrophic background was a priority. However, we have repeated the experiments using a strain containing *met17Δ* in BY4742 background (see comment#2 below).

2- Strains used in each experiment should be strictly isogenic. This is apparently not the case although difficult to figure out in several cases (see point 5 below). Using non isogenic strains may severely affect the conclusions and should be avoided. For example, in supplemental Fig. 3, the two strains BY4741 and BY4742 differ at the *LYS2* locus in addition to the *MET17* (*MET15*) locus which is the one tested in the experiment. The *lys2* auxotrophy strongly affects yeast physiology.

Response: We have repeated the experiment of supplemental Fig. 3 with a *met17Δ* mutant derived from BY4742 instead of using BY4741 (**supplemental Fig. 5**). We have also added the name of strains in the legends of figures. In the process, we have noticed a few errors and omissions in the Table listing the strains that might have led to confusion. We apologize for that.

3- Although elucidating the molecular underlying mechanisms is not in the scope of this work, it would be nice to have some experiments done under sulfur and nitrogen starvation conditions in parallel, to establish which observations are specific to sulfur starvation. This is of particular importance for Fig. 1c, Fig. 3b, supplementary Fig. 2a. Specifically, the authors should address the following questions: do the responses to these two nutritional signals share some pathway components? For example, do the sulfur regulation signals need a functional TOR pathway? Is autophagy normally induced in a *met4* mutant in response to nitrogen starvation? A few simple experiments would allow to widen the conclusions and give a preview of what is taking place (or not) at the molecular level.

Response: The mechanisms underlying autophagy induction under sulfur starvation condition were completely unknown when we started this work. Trying to unveil the complete regulatory cascade in one study would not have been realistic; therefore, after observing that *ATG* genes were upregulated upon sulfur depletion, we chose to focus first on the downstream players. We believe that the set of data that we present in this paper constitutes a consistent and coherent story already providing new mechanistic insights. We intend to focus on the upstream players in a following study that will be presented separately. Yet, we have added experiments done under nitrogen starvation conditions to

emphasize the specificity of the mechanisms involved under sulfur starvation conditions, as suggested by the reviewer. In **Fig. 2**, we show that autophagy flux dynamics following sulfur and nitrogen depletion are comparable, although not similar. More importantly, we show that *ATG* gene transcription is differentially induced upon sulfur and nitrogen depletion. In fact, only *ATG8* is strongly induced upon nitrogen depletion, the other *ATG* genes are not induced, or only weakly. Moreover in **supplemental Fig. 4**, we show that inactivation of Gcn4 or Met4 does not affect autophagy flux under nitrogen starvation. We believe that these additional experiments strengthen our study by highlighting the very specific and distinctive role of Met4 and transcription regulation under sulfur starvation. We are thankful to the reviewer for the suggestion. We have not compared the survival capacity of autophagy-deficient mutants under sulfur and nitrogen starvation. This would certainly be an interesting experiment, but difficult to interpret and we were not convinced that this comparison would provide useful information regarding the mechanisms of regulation of autophagy.

4- The role of Atg41 in the response to sulfur starvation is unclear. Data in supplementary Fig.2a show that the *atg41* mutation blocks GFP-Atg8 processing indicating a role of Atg41 in autophagy (as expected), but also blocks GFP-Atg8 expression suggesting that a functional autophagy pathway might be required for the transcriptional response to sulfur limitation to occur. This should be clarified by investigating whether other Atg mutants (affecting different steps of the autophagy process) have the same effect and whether they affect expression of other Met4 target genes (*MET* genes).

Response: We have added two sets of data in **supplemental Fig. 2**. First, we have used the Pho8 Δ 60 assay to monitor autophagy flux/activity. The results consolidate those obtained with the GFP-Atg8 processing assay and confirm that Atg41 is required for autophagy induction under sulfur starvation. Second, we have performed transcriptional analyses. We show that upon sulfur depletion *ATG8* transcription induction is diminished in *atg41* Δ compared to WT, but not in *atg1* Δ , which indicates that *ATG8* induction upon sulfur depletion does require Atg41 but not a functional autophagy pathway. In contrast, *ATG1* is induced at a slightly higher level in *atg41* Δ , and *ATG9* or *MET3* are induced at similar levels. These new data emphasize Atg41 importance for the autophagic response under sulfur starvation; however, we believe that a more detailed analysis of its precise molecular function is out of the scope of the present work and should be addressed in another study.

Minor points:

5- The name of the strains used in each figure should be clearly specified. For example, in Fig. 1c, "WT" could refer to several strains in supplementary Table1.

Response: Please see our reply to point #2.

Reviewer #2

In the current manuscript, the authors analyze the autophagy response of budding yeast upon sulfur starvation. They identify a strong transcriptional response for autophagy genes mediated by the known central transcription factor Met4 in the absence of sulfur. Atg41, a factor highly upregulated upon sulfur starvation in a Met4-dependent manner, plays a critical role for Atg8 expression and autophagy flux. The authors convincingly dissect the possible metabolic inputs for autophagy regulation and identify S-adenosylmethionine as a key metabolite.

The authors describe an interesting dataset and use systematic yeast genetics to identify the key metabolite SAM for sulfur starvation-induced autophagy. However, the study falls short in providing initial mechanistic insights into how SAM levels might be connected to the regulation of Met4 and/or autophagy and, importantly, ignores published data that have linked SAM to autophagy regulation, as described below.

We thank the reviewer for his/her appreciation.

Critical points:

(1) SAM has been linked to the regulation of autophagy in a previous publication at mechanistic level; SAM is a substrate for the methyltransferase Ppm1, which regulates the protein phosphates PPA2; methylated PPA2 promotes dephosphorylation of Npr2, a negative regulator of TORC1 and positive regulator of autophagy (Sutter et al. Cell 2013). It is inappropriate that the authors cite this article, when speculating about a link of TORC1 and Met4 regulation, but do not mention the described role of SAM for autophagy regulation. Clearly, these mechanisms might also underlie autophagy and/or Met4 regulation during sulfur starvation-induced autophagy.

Response: We apologize for having given the impression of ignoring the work published by Sutter *et al* (2013). This was not our intention. We had mentioned their publication in a more detailed and appropriate manner in the first draft of our manuscript, but the part was removed afterwards due to space constraints. We were not sure if it was pertinent to go too much into the details of this study. The regulatory mechanism described in Sutter et al (2013) is specific to a particular change of growth condition (referred to as non-nitrogen-starvation, NNS), which consists in transferring cells from rich medium (1% yeast extract, 2% peptone) to synthetic medium (0.17% yeast nitrogen base, 0.5% ammonium sulfate), with lactate as carbon source in both cases. Lactate is a respiratory carbon source, implying different metabolic and redox status compared to glucose. Moreover, NNS-induced autophagy was originally observed in CEN.PK background but not in S288C background (Wu & Tu, Mol Biol Cell 22, 4124-4133, 2011), from which BY4742 was derived. In the revised version of the manuscript, we have rewritten the sentence where Sutter *et al* (2013) is cited to clearly acknowledge

their discovery that SAM is linked to NNS-induced autophagy regulation through Ppm1 and PP2A. We have also added in Discussion a figure (**Supplementary Fig. 7**) showing that in our conditions *ATG* genes are activated and repressed normally in a *ppm1Δ* mutant following sulfur depletion/repletion, which suggest that the mechanisms involved in NNS-induced autophagy and sulfur depletion-induced autophagy are most likely different.

(2) Overall, the manuscript lacks insight into how Atg8 protein levels, autophagy flux and the transcriptional program are linked and/or their mechanistic bases. How SAM levels are translated into Met4 and/or autophagy flux are key questions.

Response: We understand the reviewer's criticism about our still incomplete understanding of the mechanisms governing autophagy regulation under sulfur starvation. However, the study presented in this manuscript is quite dense, has taken a lot of hard work and time and, above all, already provides original and novel mechanistic insights. Including the finding that the autophagic response to sulfur availability strongly relies on *ATG* gene transcriptional regulation; the link between *ATG* gene activation and induction of autophagic activity under sulfur starvation; the role of Met4 as transcriptional activator and SAM as signaling metabolite. Understanding precisely how SAM is sensed and how *ATG* gene activation ultimately translates into higher levels of autophagy are among the crucial questions that we will address in the future. Hypothesis are proposed in Discussion, that we will explore in future investigations.

(3) The essential role for Atg41 for sulfur starvation-induced autophagy is really interesting. In particular, it is interesting that Atg8 levels and turnover are significantly impaired in the absence of Atg41. Is the effect on Atg8 protein levels specific to the absence of Atg41 or does a block in any core ATGs cause these effects?

Response: We thank the reviewer for the comment. Please see our response to the point #4 of the first reviewer. We have performed Pho8Δ60 activity measures that consolidate the hypothesis that Atg41 is important for sulfur-starvation induced autophagy. In addition, we have added new data in **Supplementary Fig. 2** showing that inactivation of Atg41, but not Atg1, causes a decrease in *ATG8* transcription induction, which suggests a specific role of Atg41 in the regulation of Atg8 expression.

Reviewer #3

Summary

Prigent et al. provide evidence that the master regulator of sulfur metabolism in yeast, Met4, activates the transcription of *ATG* genes in response to sulfur starvation, and suggest that this

signaling is specifically in response to SAM depletion. The authors also demonstrate the importance of sulfur starvation-induced autophagy for cellular viability after prolonged sulfur deprivation.

Validity

In general, the interpretation of data and conclusions drawn by the authors are reasonable. While the ChIP experiments demonstrate that Met4 is recruited to the ATG genes examined, it is less obvious that Met4 is affecting the mRNA levels of these genes. There appears to be significant alteration in the levels of ATG mRNA during the time course (e.g., up 40 min, down 80 min, up again 120 min) in Fig 3A for the WT strain that make data interpretation and significance difficult to assess, in particular with respect to the *met4* and *gcn4* single deletions. Can the authors explain this?

Response: The “up-down-up” transcription pattern observed in the WT strain in the case of *ATG* and *MET3* genes is not a typical pattern (see for instance Fig.1) but is due to the SAM supplement. To address the concern about the difficulty to interpret the results owing to this pattern, we have repeated the transcription analyses and the GFP-Atg8 processing experiments of Fig. 3 using media containing 0.01 mM SAM instead of 0.05 mM (**Fig. 4** in the revised version). We find that at lower SAM concentration, transcription of *ATG* and *MET3* genes do not go down at 80 min. Moreover, *ATG* gene transcription and GFP-Atg8 expression are more diminished in *met4Δ* (comparatively to the WT cells) when the growth medium contains 0.01 mM SAM instead of 0.05 mM, which highlights better Met4 requirement. One possibility to explain the difference could be that Gcn4 expression and/or access to *ATG* gene promoters depends on SAM concentration and is higher when SAM levels are increased, resulting in higher Gcn4 occupancy at *ATG* promoter in Met4 absence of. We also observed that GFP-Atg8 processing appears more efficient in *met4Δ* grown in 0.01 mM SAM than in 0.05 mM. It could be because the impaired autophagic capacity of *met4Δ* (as shown by the Pho8Δ60 essay in **Fig. 4c**) is still sufficient to process efficiently the low GFP-Atg8 amounts in the cells grown in 0.01 mM SAM, but not when the cells are grown in 0.05 mM SAM. Since the purpose of this figure was to show the role of Met4 in autophagy regulation and not study the effect of variations in SAM concentration in the growth medium, we have chosen to keep only the results obtained in 0.01 mM SAM and remove those obtained in 0.05 mM.

Significance

The work done by Prigent et al. potentially represent a significant advance in the understanding of the mechanisms governing transcriptional regulation of autophagy. However, some of the data are not compelling and could be supported by additional experiments. Perhaps the authors might be overlooking some alternative mechanisms.

Response: We thank the reviewer for stressing out the significant advance of our study. We believe that the additional experiments we performed consolidate our conclusion and make less likely the possibility of alternative mechanisms.

Data and Methodology

The figures and data are organized and presented clearly. The utilization of specially formulated “sulfur free” media is a strength of their methodology, as well as the use of specific sulfur metabolism mutants to identify the responsible metabolite. However, one concern the authors should look into is the use of BY strain background that may be auxotrophic for *met15Δ*, which could alter the response to sulfur and methionine metabolism or synergize with *met4Δ* deletion.

Response: All the strains, except for the experiment in **Supplementary Fig. 5**, contain the wild-type *MET15* allele (named *MET17* in the manuscript to follow the standard nomenclature). We apologize for not indicating in the previous version of our manuscript the name of the strains in the figure legends. We have rectified the omission in the revised version. As mentioned by the reviewer, the presence of *met17Δ* would have been a real concern.

They should consider validating key results using a prototrophic strain or other strain background such as W303.

Response: Several experiments were repeated using a prototrophic strain. Please see our response to the first comment of the first reviewer.

Analytical Approach

The authors should state the n for their RT-qPCR experiments, unless I have missed it.

Response: We have rectified the omission in the revised version.

Suggested Improvements

Major points:

The authors rely heavily on the GFP-Atg8 cleavage assay to monitor autophagy. While some results obtained with this assay are consistent with their model, the marked reduction in the levels of GFP-Atg8 protein in the *met4* and *gcn4* single and double mutants compared to WT makes it difficult to assess levels of autophagy by the very nature of the mutants they are testing. This also applies to their microscopy data and other data where the growth condition alters the abundance of GFP-Atg8. While this reduction in GFP-Atg8 is informative in and of itself, the use of a complementary and more quantitative method to monitor autophagy such as the Pho8 60 alkaline phosphatase assay would strengthen the work. After all, the induction of ATG genes and protein amounts may not matter in

the end if ATG8 is not processed, etc. and there is comparable delivery of autophagosomes to the vacuole. To this point, the extent of processing of ATG8 for various mutants does not appear that different, when it might be expected that methionine addition should repress actual autophagy (e.g., Fig 5b, 5e) – these observations would appear inconsistent with the authors initial finding that sulfur starvation induces autophagy.

Response: As suggested by the reviewer, we have added *Pho8Δ60* activity measures in two figures: the figures with the *met4Δ* and *gcn4Δ* mutants, and the figure with the *atg41Δ* mutant (**Fig. 4** and **Supplementary Fig. 2**, respectively). The results showed in both cases a decreased of Pho8Δ60 activity in the mutants compared to WT, confirming that autophagic activity is strongly reduced, which correlates with the decrease of GFP-Atg8 and free GFP levels.

Concerning the GFP-Atg8 assay, we agree that the percentage of free GFP over total GFP may be more difficult to interpret. To highlight the differences in the amounts of free GFP between the strains and starvation conditions, graphs giving the levels of GFP release were systematically added alongside the graphs giving the levels of GFP-Atg8 expression and processing. In Fig. 5b and 5e (**Fig. 6b** and **6e** in the revised version) we observed high expression and efficient processing of GFP-Atg8 in the two mutants upon transfer into the medium completely deprived of sulfur, demonstrating strong autophagy induction. However, when the mutants are deprived of cysteine only (in the case of *cys4Δ*) or methionine only (in the case of *met6Δ mht1Δ sam4Δ*), GFP-Atg8 expression is lower compared to total sulfur depletion, but not completely shut down, and the low amount of GFP-Atg8 expressed can be processed fairly efficiently. However, the levels of free GFP release are significantly reduced, meaning lower autophagy levels, in the conditions where only cysteine or methionine are depleted. The Discussion section has also been rewritten (see the second paragraph in particular) to better highlight this point. Altogether, the two assays support the model that autophagy induction in response to sulfur depletion results mainly from SAM depletion.

The authors provide evidence that Met4 may bind several ATG genes using CHIP, however the prominent induction of ATG genes can still be observed in the absence of Met4. This is quite confusing and suggests that the observed Met4 binding may not be functionally relevant. If Met4 were indeed a regulator of ATG genes, one might expect effects on mRNA levels similar to the extent observed with the MET3 transcript.

Response: The experiments with the *met4Δ* and *gcn4Δ* single and double mutants were repeated in slightly different growth conditions (see the reply to the Validity point) and we believe that the results demonstrate more convincingly the role of Met4 in the transcriptional regulation of ATG genes. Even if Met4 is a regulator of ATG genes, we do not think that ATG genes should obligatory be

affected in the *met4Δ* mutants in the same manner as *MET* genes. Autophagy can be induced in response to a variety of cellular challenges and signals; therefore, one can suppose that the regulation of *ATG* genes should be more complex and involve a larger variety of regulators than the regulation of *MET* genes, including regulators that could be also responsive to sulfur depletion but not bind *MET* genes.

Can the authors also observe binding of Cbf1, Met31 or Met32?

Response: We have not performed CHIP experiments on Cbf1, Met31 and Met32. This type of experiments is quite demanding and time-consuming, and was not considered as a decisive priority.

Although significant effects were observed in the double *gcn4 met4* knockout, there is the concern that this strain could be quite sick, as *Gcn4* is generally required for transcriptional response to amino acid starvation.

Responses: In fact, the *gcn4Δ* mutation has not much effect on cell growth when a full complement of amino acids is supplied. The *met4Δ* mutation is more impacting. In SF-medium supplemented with 0.1 mM methionine/0.01 mM SAM, the WT and *gcn4Δ* strain have a doubling time of around 1h40 min at 30°C, the *met4Δ* mutant around 2h, and the double mutant around 2h30. Therefore, the *gcn4Δ met4Δ* mutant is definitely sick but still grows relatively well when amino acids and SAM are provided. Note that the *sam1Δ sam2Δ* mutant does not grow better but is still capable of inducing *ATG* genes and autophagy upon sulfur depletion.

Minor points:

The authors should consider including growth rates of various mutants under normal versus starvation conditions.

Response: We do not see where to integrate growth rates in the manuscript and what would be the justification. We are not sure it would be comprehensible without the need to address a specific question. All the mutants that we used are viable. Our synthetic complete medium contains all that is required to support efficient fermentative growth including all amino acids and the two nucleotides adenine and uracil. The only mutant that may be cause for concern is *met4Δ*. Met4 inactivation decreases expression of SAM synthetase genes, leading to cellular SAM levels insufficient to support normal growth; however, the resulting slow growth phenotype can be rescued by adding SAM in the medium. The matter has been studied in details in Hickman et al, Mol. Biol. Cell 22: 4192-4204, 2011. After sulfur depletion, the WT cell population keep growing a little, achieving between 1 and 1.5 doubling and then stop growing. In contrast, the cells containing *met4Δ* stop growing before the

population can double, probably because impaired autophagy response and lower intracellular sulfur pools. These aspects are visible in the cell viability experiments **Fig. 1c** and **Supplementary Fig. 3c**.

Grammar and spelling on lines 55 (called the autophagosome), 70 (diverse), 111 and 310 (let hypothesize)

Response: Grammar and spelling mistakes were corrected.

REVIEWER COMMENTS

Reviewer #1 (Remarks to the Author):

My concerns have been addressed in the revisions

Reviewer #2 (Remarks to the Author):

The authors addressed my concerns.

Reviewer #3 (Remarks to the Author):

The authors have included key additional experiments involving prototrophic strains and an additional assay for autophagy (alkaline phosphatase reporter). I am largely satisfied with how they addressed my specific comments:

Validity:

"Up-down-up" transcription pattern: the authors have addressed this issue by using a lower SAM concentration for their treatments (0.01 vs 0.05 mM). It could be reassuring if the authors performed additional autophagy experiments showing that treatment with 0.01 mM SAM can repress autophagy using the alkaline phosphatase assay and/or another GFP cleavage assay.

Significance:

Authors say their new experiments address this issue.

Data and Methodology:

Authors have repeated certain experiments with prototrophic strains. Interpreting which experiments were performed with BY or prototrophic yeast remains difficult. Authors should clearly highlight which experiments were performed with a given strain in the figures if possible.

Analytical Approach:

Authors addressed issue.

Suggested Improvements:

Major Points:

GFP-Atg8 assays:

The authors have addressed the concern over the GFP-Atg8 assays by performing two alkaline phosphatase assays and adding graphs to accompany the GFP-Atg8 assays. I believe the GFP-Atg8 assay remains a problematic assay for monitoring autophagy flux but can represent the abundance of Atg8 which the authors propose is dependent on Met4 and SAM levels along with the levels of other Atg proteins. In my opinion, it remains incumbent on the authors to better show how autophagy flux can change in their mutant cells and various media conditions. The authors could perform additional alkaline phosphatase assays and/or perform additional GFP cleavage assays (ie Idh1-GFP cleavage assays) to better assess autophagy flux.

ChIP issues:

The authors reasonably propose that ATG transcription may not be entirely dependent on Met4 but could be regulated by other factors. It remains poorly understood whether Met4 is actively inducing ATG gene transcription in a meaningful way to regulate autophagy or if the effects of Met4 on autophagy are primarily posttranscriptional. After all, autophagy is apparently highly repressed in met4-ko cells while ATG transcript levels remain elevated.

The authors declined to further assess Cbf1, Met31, Met32 binding because it would be a lot of work.

Specific comments:

The use of strains of different backgrounds makes interpretation of the data confusing. Genetic backgrounds could be depicted in the figures to make interpretation easier.

The GFP-Atg8 assays are difficult to interpret in certain cases in which the GFP-Atg8 abundance is diminished in the mutants. The use of another GFP cleavage assays (e.g., Idh1-GFP cleavage) could further support their claims that sulfur starvation dependent autophagy is dependent on Met4 and SAM levels.

Have the authors assessed what happens to ALP activity when SAM is replenished in sam1sam2 DKO cells to make the point that autophagy flux and ATG protein levels can be repressed with SAM. Have the authors tested how autophagy is induced by SAM depletion using an assay other than GFP-Atg8 cleavage?

ChIP experiments for co-factors (Cbf1, Met31, Met32) is a reasonable request that could substantially strengthen the findings. I don't really agree with their rebuttal responses on this and on the growth curves.

Typo - Line 200 - 'not significant' needs to be changed to 'no significant'?

One thing that remains puzzling is that the levels of the different ATG mRNAs actually start off at higher amounts in the met4 mutant compared to WT. There could be a disconnect here between what is happening at the transcriptional level and what is happening at the protein level. For example, if ATG1 transcript levels are ~4-fold higher at t=0 in the met4

mutant, does it matter if transcript levels do not increase to the extent seen in WT at later time points? Levels of Atg1 protein could be comparable between WT and the mutant during sulfur starvation if the mutant appears to have a “head start”. Have the authors assessed protein levels of Atg1, Atg9, and Atg41 in WT vs met4 KO to justify the conclusion that the reduced transcriptional induction actually results in lower levels of these proteins, and presumably the lower autophagy flux?

Responses to reviewers' comments on manuscript NCOMMS-22-17321

Reviewers' comments are in blue; responses are in black. Addition introduced in the manuscript itself are marked in dark red.

Reviewer #1:

My concerns have been addressed in the revisions.

Reviewer #2

The authors addressed my concerns.

Reviewer #3

The authors have included key additional experiments involving prototrophic strains and an additional assay for autophagy (alkaline phosphatase reporter). I am largely satisfied with how they addressed my specific comments:

Validity:

"Up-down-up" transcription pattern: the authors have addressed this issue by using a lower SAM concentration for their treatments (0.01 vs 0.05 mM). It could be reassuring if the authors performed additional autophagy experiments showing that treatment with 0.01 mM SAM can repress autophagy using the alkaline phosphatase assay and/or another GFP cleavage assay.

Re: Fig. 4c shows that *pho8Δ60* ALP activity is repressed in all strains, including *met4Δ* and *gcn4Δ met4Δ*, in growth medium containing 0.01 mM SAM & 0.1 mM methionine (see the zero time point before transfer to depleted medium). It should be reminded that *met4Δ* still possesses some SAM synthetase activity allowing conversion of methionine into SAM. *Met4* inactivation causes a decrease in SAM1 and SAM2 expression, not a total arrest, causing a slow growth phenotype that can be rescued by adding small amounts of SAM in the medium (see for instance Hickman et al., Mol Biol Cell 22:4192-204, 2011). The question of knowing how much intracellular SAM is necessary to repress autophagy is interesting but quite difficult to address. Metabolomic analysis would be necessary to address precisely this question.

Significance:

Authors say their new experiments address this issue.

Data and Methodology:

Authors have repeated certain experiments with prototrophic strains. Interpreting which experiments were performed with BY or prototrophic yeast remains difficult. Authors should clearly highlight which experiments were performed with a given strain in the figures if possible.

Re: We believe that adding strain genotypes in the figure would complicate their reading. The legends contain all information regarding the strains and the use of prototrophs is mentioned whenever necessary. To address the reviewer's comment, we have reformulated the first sentence of "Yeast strains and growth media" in "Methods" section to better stress out that our experiments used auxotrophic strains unless otherwise stated in the text and figure legends.

Analytical Approach:

Authors addressed issue.

Suggested Improvements:

Major Points

GFP-Atg8 assays: The authors have addressed the concern over the GFP-Atg8 assays by performing two alkaline phosphatase assays and adding graphs to accompany the GFP-Atg8 assays. I believe the GFP-Atg8 assay remains a problematic assay for monitoring autophagy flux but can represent the abundance of Atg8 which the authors propose is dependent on Met4 and SAM levels along with the levels of other Atg proteins. In my opinion, it remains incumbent on the authors to better show how autophagy flux can change in their mutant cells and various media conditions. The authors could perform additional alkaline phosphatase assays and/or perform additional GFP cleavage assays (ie Idh1-GFP cleavage assays) to better assess autophagy flux.

Re: We have added an additional GFP cleavage assay, the Pgc1-GFP cleavage assay, to complement the experiments with the metabolic mutants *cys4Δ*, *met6Δ mht1Δ sam4Δ* and *sam1Δ sam2Δ* (new Supplementary Fig. 8). The new data consolidate and extend our previous results. They first show that Pgc1-GFP is cleaved in an autophagy-dependent manner following sulfur depletion. They also show that methionine depletion in *met6Δ mht1Δ sam4Δ* induces very weakly Pgc1-GFP cleavage, whereas depletion of SAM in *sam1Δ sam2Δ* induced Pgc1-GFP cleavage almost as efficiently as sulfur depletion. These two results are in line with the results of the GFP-Atg8 cleavage assay. By contrast, cysteine depletion in *cys4Δ* induces higher levels of cleaved Pgc1-GFP than might be expected based on the GFP-Atg8 cleavage assay. In fact, in this mutant, levels of cleaved Pgc1-GFP are similar upon cysteine and sulfur depletion while levels of GFP-Atg8 expression and GFP release are lower upon cysteine depletion compared to sulfur depletion. One hypothesis could explain these seemingly contradictory results: cysteine depletion signals the induction of bulk autophagy, not induction of selective autophagy. By contrast, sulfur depletion, which also causes SAM depletion, would signal the induction of both types of autophagy. One functional consequence would be that the type of cargoes sequestered by autophagosomes might vary depending on the sulfur compound.

The text of the manuscript was modified in the “Results” and “Discussion” sections to integrate the new results.

ChIP issues: The authors reasonably propose that ATG transcription may not be entirely dependent on Met4 but could be regulated by other factors. It remains poorly understood whether Met4 is actively inducing ATG gene transcription in a meaningful way to regulate autophagy or if the effects of Met4 on autophagy are primarily posttranscriptional. After all, autophagy is apparently highly repressed in *met4*-ko cells while ATG transcript levels remain elevated. The authors declined to further assess Cbf1, Met31, Met32 binding because it would be a lot of work.

Re: We believe that our ChIP experiments provide good evidence that Met4 associates with ATG genes. The fact that the promoters contain binding sites for Met31/Met32 (and Cbf1 in some cases) gives additional support to the hypothesis that Met4 actively induces their transcription. Met4 is a transcriptional activator possessing a characteristic activation domain and a bZIP DNA-binding domain. There is no data suggesting that Met4 might have other functions that would support the possibility of a posttranscriptional role. Nevertheless, we cannot completely rule out the possibility of an indirect role involving its transcriptional activation and DNA-binding capacities, but performing only additional ChIP with Cbf1, Met31 or Met32 will not answer the question.

Specific comments

The use of strains of different backgrounds makes interpretation of the data confusing. Genetic backgrounds could be depicted in the figures to make interpretation easier.

The GFP-Atg8 assays are difficult to interpret in certain cases in which the GFP-Atg8 abundance is diminished in the mutants. The use of another GFP cleavage assays (e.g., Idh1-GFP cleavage) could

further support their claims that sulfur starvation dependent autophagy is dependent on Met4 and SAM levels.

Please see our responses above.

Have the authors assessed what happens to ALP activity when SAM is replenished in *sam1sam2* DKO cells to make the point that autophagy flux and ATG protein levels can be repressed with SAM.

Re: The paper already contains a great amount of data. Its primary object is the transcriptional regulation of autophagy. In this regard, the aim of the depletion-repletion experiments in Fig. 9 is to strengthen the role of SAM in the regulation of *ATG* genes. The question of knowing how autophagy is repressed when cells are replenished with sulfur following a period of starvation is important and should be investigated in details in a separate study.

Have the authors tested how autophagy is induced by SAM depletion using an assay other than GFP-Atg8 cleavage?

Re: We have performed additional experiments using the Pgc1-GFP cleavage assay to monitor autophagy. Please see our response above.

ChIP experiments for co-factors (Cbf1, Met31, Met32) is a reasonable request that could substantially strengthen the findings. I don't really agree with their rebuttal responses on this and on the growth curves.

Re: We have now added several growth curves (new Supplementary Fig. 8). They show that the metabolic mutants do not have major growth defects in replete medium whereas they stop growing in depleted medium. They also show that WT and metabolic mutants grow no more than one generation upon transfer into depleted medium, while they keep growing for 2.5 to 3 generations upon transfer in replete medium, which confirms that the growth arrest is due to the depletion of the sulfur compound.

Typo - Line 200 - 'not significant' needs to be changed to 'no significant'?

Re: The mistake was corrected.

One thing that remains puzzling is that the levels of the different *ATG* mRNAs actually start off at higher amounts in the *met4* mutant compared to WT. There could be a disconnect here between what is happening at the transcriptional level and what is happening at the protein level. For example, if *ATG1* transcript levels are ~4-fold higher at t=0 in the *met4* mutant, does it matter if transcript levels do not increase to the extent seen in WT at later time points? Levels of Atg1 protein could be comparable between WT and the mutant during sulfur starvation if the mutant appears to have a "head start".

Have the authors assessed protein levels of Atg1, Atg9, and Atg41 in WT vs *met4* KO to justify the conclusion that the reduced transcriptional induction actually results in lower levels of these proteins, and presumably the lower autophagy flux?

We do not have explanation for the higher *ATG1*, 8 and 41 mRNA levels observed at t0 in *met4Δ*. We have not quantified the protein levels of Atg1, 8, 9 or 41, but Fig. 4b and Supplementary Fig. 3b show that GFP-Atg8 levels at t0 are comparable between WT and *met4Δ* (difference is less than twofold). It is difficult to speculate how much effect could have these higher mRNA levels at the time of depletion on the dynamics of the autophagic response of *met4Δ*. Nevertheless, the results of the GFP-Atg8 and Pho8D60 assays demonstrate a clear decrease in autophagy in *met4Δ*.